# HiBO: Hierarchical Bayesian Optimization via Adaptive Search Space Partitioning

## Abstract

Optimizing opaque functions in high-dimensional search spaces has been known to be challenging for traditional Bayesian Optimization (BO). In this paper, we introduce HiBO, a novel hierarchical algorithm integrating global-level search space partitioning information into the acquisition strategy of a local BO-based optimizer. HiBO employs a search-tree-based global-level navigator to adaptively split the search space into partitions with different sampling potential. The local optimizer then utilizes this global-level information to guide its acquisition strategy towards the most promising regions within the search space. A comprehensive set of evaluations demonstrates that HiBO outperforms state-of-the-art methods in high-dimensional synthetic benchmarks and presents significant practical effectiveness in the real-world task of tuning configurations of database management systems (DBMSs).

## 1 Introduction

Bayesian Optimization (BO) has emerged as a powerful approach for optimizing ***opaque functions*** that exhibit little information of its form and requires iterative evaluations for reaching the optima with significant computational cost. BO has been applied to a wide range of real-world scenarios requiring optimization over opaque functions within a limited budget of sampling times, such as Neural Architecture Search (Kandasamy et al., 2018; Ru et al., 2020), hyperparameter tuning (Hvarfner et al., 2022; Zimmer et al., 2021) and automatic database management system (DBMS) configuration tuning (Van Aken et al., 2017; Zhang et al., 2021; 2022). Typically, BO iteratively builds a probabilistic surrogate model on the objective function and uses an acquisition function to decide where to sample next, balancing exploration and exploitation to efficiently locate the optimum. However, it is well known that BO struggles to scale in high-dimensional search spaces (Binois & Wycoff, 2022; Malu et al., 2021). As dimensionality increases, the size of the search space grows exponentially and uniformly sampled points are more likely to be distant from each other (Rashidi et al., 2024). Such sparsity of nearby data increases uncertainty in the posterior distribution, which can cause the acquisition function to favor unexplored regions excessively, leading to over-exploration.

Even worse, optimization over high-dimensional search spaces in real-world systems is more challenging. The task of DBMS configuration tuning exemplifies this challenge, which involves a high-dimensional configuration knob space, complex interdependencies between knobs, and noisy performance observations (Zhao et al., 2023). Existing approaches towards this problem either rely on extensive pre-collected random samples (Van Aken et al., 2017) or metadata of prior workloads (Zhang et al., 2021), or restrict evaluations to manually selected low-dimensional spaces (Zhang et al., 2022; Cereda et al., 2021). Moreover, methods aiming to scale BO to high dimensionalities (to be discussed below) often lack evaluation on real-world systems, limiting insights into their practical effectiveness.

Various approaches have been proposed towards mitigating the curse-of-dimensionality. Some rely on additive models applied to separate groups of dimensions (Duvenaud et al., 2011; Kandasamy et al., 2015), while others project the high-dimensional input space into a lower-dimensional subspace using linear transformations (Wang et al., 2016; Letham et al., 2020; Nayebi et al., 2019). But these methods assume specific structural properties, such as decomposability of the objective function or the existence of a hidden low-dimensional active subspace. Another line of work avoids these assumptions by restricting the sampling scope into a promising region at each iteration. Among them, some define this region as the trust region (TR) maintained around the best samples found so

far (Eriksson et al., 2019; Daulton et al., 2022) while others partition the search space and focus on the most promising regions (Wang et al., 2020; Munos, 2011; Bubeck et al., 2011; Kim et al., 2020). However, these methods greedily confine the sampling scope to the only region with highest immediate potential, underutilizing the structural information generated during the partitioning process, such as the way the search space is partitioned. Such inefficient utilization of generated structural information and inflexible sampling scope confinement can lead to suboptimal performance. Moreover, with a limited initial sample budget, the constructed tree may lack reliability, and strict confinement can further prevent effective exploration of the search space. Our experiments empirically demonstrate the suboptimal performance of such approaches, where our proposed method offers ∼28% performance improvement in real-world systems than them. While incorporating such structural information into the original BO algorithm for optimal performance introduces challenges, including designing an effective hierarchy to combine the extra module extracting structural information with the original BO, and ensuring that the structural information are utilized optimally. On the other hand, the way of partitioning the search space is critical in terms of exploitation-and-exploration trade-offs and the computational cost. Splitting the search space into coarser or fine partitions directly impacts the generated structural information and the computational overhead in this process. An adaptive mechanism for adjusting the partitioning process is needed.

Motivated by the limitations of existing high-dimensional BO methods and the need for effective real-world system tuning, we have the following contributions:

**Hierarchical Framework for Bayesian Optimization**. We propose **Hi**erarchical **B**ayesian **O**ptimization (HiBO), a hierarchical BO variant that is the first to integrate global-level space partitioning information into the local BO model's acquisition strategy instead of rigidly confining the sampling scope. HiBO is shown to provide ∼28% more performance improvements than only using the local optimizer in high-dimensional DBMS configuration tuning tasks with limited sample budget.

**Search Space Partitioning with Adaptive Control**. We introduce the search-tree-based space partitioning for the global-level navigator, recursively splitting the search space into the partitions with different sampling potential. Besides, an adaptive mechanism on controlling the maximum search tree depth is proposed for effective exploration-and-exploitation trade-offs and computational cost reduction.

**Practical Evaluation on Real-world Benchmark**. Besides experiments on synthetic and simulated "real-world" benchmarks (Jones, 2008; Wang et al., 2017), we demonstrate HiBO's effectiveness and practical value via evaluation on a real-world DBMS configuration tuning task, comprehensively considering performance improvement, tuning time cost and failure rate.

## 2 BACKGROUND

### 2.1 PRELIMINARIES

Bayesian Optimization (BO) is an effective technique for optimization that iteratively refines a probabilistic surrogate model of the objective function and employs an acquisition function to determine the next sampling points (Candelieri, 2021).

The goal is to solve the following problem:

$$x^* = \underset{x \in X}{arg\,max}\ f(x) \tag{1}$$

where $x$ is typically a data point in $d$-dimensional search space $X \subset \mathbb{R}^d$ and $f(x)$ is a opaque function mapping the input data into the performance metric of interest. This function is typically computationally expensive to evaluate and provides limited explicit information, hence the term "opaque".

A typical BO-based algorithms begins with an initial random sampling of data points. In each iteration of its optimization loop, it fits a surrogate model to estimate the posterior probability distribution over the objective function based on the collected data. It then selects the next point maximizing the acquisition function, evaluates this point by the objective function and updates the history dataset. As iterations progress, the surrogate model becomes increasingly accurate at estimating the distribution of the objective function, leading to better sampling decisions.

|  | HOO | LA-NAS | LA-MCTS | **HiBO** |
|---|---|---|---|---|
| **Partitioning Op.** | K-nary | LR | Clustering & Classification | Clustering & Classification |
| **Depth Control** | × | × | × | Adaptive Control |
| **Restricted Sampling** | Yes | Yes | Yes | No |
| **Sampling Strategy** | Random | Random/BO | BO | BO with augmented acqf. |

**Table 1:** Comparison between optimization algorithms based on space partitioning, where *LR* refers to Linear Regression and *acqf.* refers to acquisition function. *Partitioning Op.* is the operation for splitting the search space (see Section 4.1 for more details).

Though BO has been recognized as a powerful approach across various problem domains, vanilla BO is known to be inherently vulnerable to the "curse-of-dimensionality" (Binois & Wycoff, 2022) when the search space has a large number of dimensions. As dimensionality increases, the average squared distance between uniformly sampled points grows, making it harder for distance-based surrogate models to capture correlations between data points (Rashidi et al., 2024). Additionally, the increased distance can result in greater uncertainty assigned to data points by the estimated posterior, increasing acquisition functions unreasonably and trapping BO algorithms in cycles of excessive exploration ("over-exploration"; Siivola et al. (2018)).

## 2.2 RELATED WORK

Several paradigms of approaches have been developed to scale Bayesian Optimization (BO) to high-dimensional search spaces. As discussed in Section 1, some methods leverage structural assumptions about the objective function, such as additive kernels (Duvenaud et al., 2011; Kandasamy et al., 2015; Wang et al., 2018) or subspace embeddings (Wang et al., 2016; Letham et al., 2020; Nayebi et al., 2019; Papenmeier et al., 2022). Alternatively, variable-selection-based strategies are employed to identify the most important dimensions (Li et al., 2017; Spagnol et al., 2019; Shen & Kingsford, 2023; Song et al., 2022). However, these methods often rely on assumptions of the input space that may not always hold in practice, or involve uncertain user-estimated parameters for these assumptions, such as the existence of the latent subspace and its dimensionality respectively.

Another line of work focuses on optimizing the sampling scope without such assumptions, which can be further divided into 1) local modeling (Eriksson et al., 2019; Daulton et al., 2022) and 2) space partitioning (Munos, 2011; Bubeck et al., 2011; Wang et al., 2021; 2020). TuRBO, a representative of local modeling, maintains trust regions around the best samples to mitigate over-exploration. While prior space-partitioning-based methods focus on splitting the search space into different partitions and limiting the sampling scope to the most promising one. A detailed comparison space-partitioning-based optimization is presented in Table 1, which includes HOO (Bubeck et al., 2011), LA-NAS (Wang et al., 2021), LA-MCTS (Wang et al., 2020) and HiBO. HOO traditionally applies axis-aligned K-nary partitioning for splitting the space while LA-NAS extends it by leveraging linear regression to learn the linear decision boundary for bifurcation. As a more recent work, LA-MCTS applies MCTS-inspired ideas to partition the search space based on clustering and classification for more flexible space bifurcation, and becomes the state-of-the-art (SOTA) approach among space-partitioning-based work. In our work, HiBO extends these MCTS-based concepts in LA-NAS and LA-MCTS by formalizing the notion of sampling potential and employing a general search tree instead of MCTS. HiBO's partitioning search tree can be adaptively adjusted during the search to balance exploration-exploitation trade-off, and reduce unnecessary computational cost. Additionally, HiBO does not restrict sampling scope to the only specific region, allowing greater flexibility and more comprehensive consideration collected samples.

It is important to distinguish our use of the term "Hierarchical Bayesian Optimization" from its usage in Pelikan et al. (2007). While Pelikan et al. (2007) also employs this term, their discussions focus on scalable solutions for nearly decomposable and hierarchical problems. While HiBO does not assume any decomposability or hierarchical properties in the optimization problems it addresses. The "hierarchicalness" of HiBO mainly lies in the methodology, instead of the target problem. As our focus is scaling up BO to high-dimensional search space, no further discussion about their method is provided considering no direct relevance.

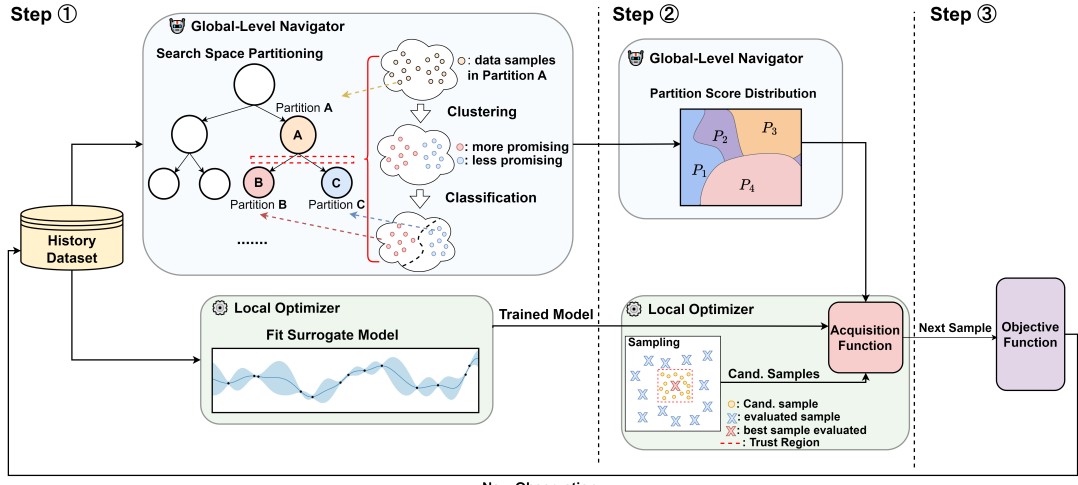

**Figure 1:** Illustration of the high-level workflow of HiBO within each optimization iteration.

## 3 SYSTEM OVERVIEW

In this section, we will introduce the high-level design of HiBO. HiBO is a **hierarchical** BO approach consisting of two parts: 1) a **global-level navigator**, which conducts search-tree-based search space partitioning based on collected data with dynamic and adaptive constraints posed when constructing the search tree; 2) a **local optimizer** with a BO-based algorithm as its core. Specifically we choose TuRBO (TuRBO-1; Eriksson et al. (2019)) as the local optimizer throughout this paper considering its practical effectiveness and simplicity for implementation.

Before the optimization loop starts, similar to typical BO-based algorithms, HiBO collects an initial set of data points (*history dataset*) based on random sampling, ensuring the basic coverage of the search space. The initial dataset allows the global-level navigator to construct the initial search tree and enables the local optimizer to train the initial surrogate model. After the initial sampling, HiBO enters the main optimization loop. During each iteration of the loop, the global-level navigator partitions the search space based on the history dataset, and the local optimizer integrates the global-level information into its acquisition strategy for biasing the sample selection towards data points from most promising regions. The high-level workflow for each iteration is outlined below, where Figure 1 illustrates this procedure:

- **Step ①**: Based on the history dataset, the global-level navigator constructs the search tree to partition the search space and generates a distribution of partition scores. While the local optimizer trains its surrogate model. The details of search space partitioning is explained in Section 4.1.

- **Step ②**: The local optimizer randomly samples a set of candidate points within the trust region and calculates the modified acquisition function values based on the trained surrogate model and partition-based scores.

- **Step ③**: The candidate with the highest adapted acquisition function value is selected for objective function evaluation, and the new observation is added to the history dataset.

The key components of HiBO mentioned in the procedure above will be explained in the following sections.

## 4 GLOBAL-LEVEL NAVIGATOR: ADAPTIVE SEARCH SPACE PARTITIONING

This section presents the design of our global-level navigator, which recursively and adaptively partitions the search space with machine learning techniques and evaluates the sampling potential of partitions balancing exploration and exploitation. We intuitively describe the ***sampling potential*** of a search space partition as the potential of sampling high-performing data points within this partition in the following sections.

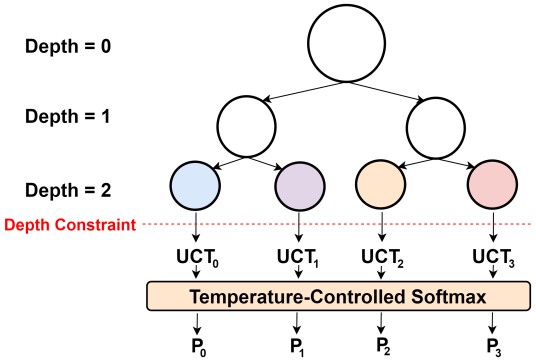
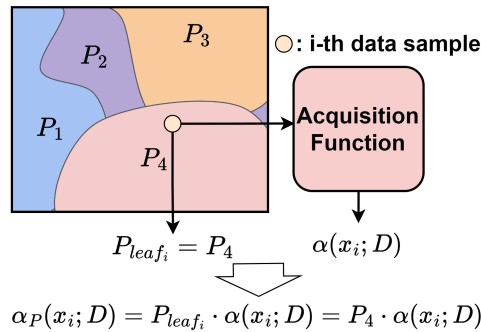

**(a)** Calculation of partition scores from an example search tree.

**(b)** Calculation of the weighted acquisition function value corresponding to a sample point (see Section 5).

**Figure 2:** Illustration of how the constructed search tree is integrated into the local optimizer's acquisition strategy.

The core motivation of introducing this global-level navigator is to partition the search space in a way maximizing the difference in sampling potential across resulting partitions. It generates a distribution of sampling potential over these partitions, which will be utilized by the local optimizer for augmenting its acquisition strategy to boost its optimization efficiency in high-dimensional search spaces (see Section 5). Besides the space partitioning itself, the design of the global-level navigator includes mechanisms for reducing computational cost and balancing trade-offs between exploration and exploitation.

### 4.1 DATA-DRIVEN SEARCH-TREE-BASED SPACE PARTITIONING

The navigator constructs a search tree in a data-driven manner, where each node represents a partition of the original search space and is associated with the samples falling into this partition. Starting from the root node representing the entire search space, this method recursively splits the space into smaller and more specific partitions, as illustrated in **Step ①** of Figure 1.

To partition a node, clustering techniques like K-Means (MacQueen et al., 1967) are used to group the samples from the history dataset that fall within this node's partition into two clusters, assigning each sample a binary label. Given each sample has its cluster label, a classification model is trained on the sample features to predict the cluster labels. This classifier hence learns a decision boundary dividing the current search space partition into two parts. Each of these two parts thus forms a bifurcation of the original partition.

By bifurcating the original partition into two sub-partitions, two child nodes are generated, each corresponding to one of the sub-partitions and containing the samples that fall within them. Recursively repeating this ML-based bifurcation starting from the root node will finally result in a binary tree of nodes, where each node represents a progressively refined region of the search space.

A key aspect of this approach is that clustering is performed on both sample features and their performance data. As a result, samples in the two resulting sub-partitions are expected to differ significantly in terms of both performance and feature similarity, enabling the differentiation of valuable data points within different partitions. Once the tree is fully constructed, each leaf node corresponds to a sequence of split operations, and represents a highly specific partition of the original search space. These refined partitions provide global-level insights into which regions of the search space are likely to be valuable for further sampling.

### 4.2 SAMPLING POTENTIAL EVALUATION VIA UCT

Once the search tree is constructed, HiBO utilizes Upper Confidence bounds applied to Trees (UCT; Kocsis & Szepesvári (2006)) to evaluate each node's sampling potential by balancing exploitation and exploration:

$$UCT_j = \hat{v}_j + 2C_p\sqrt{2\log n_p/n_j} \tag{2}$$

Here, $C_p$ is a hyperparameter controlling the exploration-exploitation trade-off, $n_j$ and $n_p$ denote the number of visits to node $j$ and its parent node respectively, and $\hat{v}_j$ represents the average objective value within node $j$. By integrating UCT, HiBO prioritizes data points from partitions with higher average performance while still exploring less-visited regions of the search space, striking a balance between exploitation and exploration.

In summary, the essence of HiBO's search-tree-based partitioning strategy involves: 1) utilizing machine learning techniques to partition the search space, aiming to maximize the difference of sampling potential across partitions; and 2) applying UCT to quantitatively assess such potential of each partition while balancing the exploration and exploitation.

### 4.3 Dynamic and Adaptive Search Tree Construction

In addition to the procedures described for search space partitioning, we address two additional concerns regarding the proposed strategies:

- **Computational Cost**. Building a search tree at each iteration can be computationally expensive, where multiple classifiers such as Support Vector Machines (Boser et al., 1992) need to be trained and are used for inferring labels of thousands of candidate data points.

- **Balance of Exploration and Exploitation** The depth of the tree, determined by the number of bifurcations from root to leaf, affects the range of the search space represented by each leaf node. As the tree deepens, leaf nodes represent smaller and more specific partitions, potentially leading to insufficient exploration.

The two concerns necessitate a dynamic and adaptive strategy to control the maximum depth of the constructed search tree during optimization. Towards this target, inspired by TuRBO's approach (Eriksson et al., 2019) for exploration control via dynamic trust region adjustment, HiBO allows the tree to expand or shrink based on consecutive successes or failures. Consecutive successes reduces the number of bifurcation and hence increases the range covered by each leaf node, thereby promoting more exploration by considering a broader set of samples with equal partition-score weights.

The additional rules for building the tree are explained as follows:

**Rule 1** (Breadth-First Search Tree Construction) Strictly follow the breadth-first order to build the tree layer by layer. Track the depth of each newly split child node and stop splitting once they reaches the current maximum tree depth, which is shown by the red line in Figure 2a.

**Rule 2** (Adaptive Maximum Tree Depth) After a certain number of consecutive successes (improvements over the current best), reduce the maximum tree depth to broaden the range covered by each leaf node. Conversely, increase the maximum depth after consecutive failures for more focused biasing and hence more exploitation.

**Rule 3** (Restart When Too Deep) If the maximum tree depth goes beyond a predefined threshold, instruct the local optimizer to restart with a new round of initial sampling and reset the maximum tree depth.

This dynamic and adaptive control on the maximum depth of the search tree aims to prevent unnecessary tree growth, reducing computational cost while maintaining a balance between exploration and exploitation. A further validation of the effectiveness of such adaptive tree depth settings are shown in Appendix C.

## 5 Local Optimizer: Partition-Score-Weighted Acquisition

This section presents the design for the local optimizer to integrate the global space partitioning information into its acquisition strategy, with key operations illustrated in Figure 2.

Given all leaf nodes represent the entire search space, each leaf node serves as an information source of the global partitioning. The UCT score of each leaf node reflects the sampling potential of each partition of the whole search space and should be effectively utilized. As illustrated in Figure 2a, after the tree is built, each leaf node $j$ is assigned with a UCT score $UCT_j$ following Equation 2. To scale all scores into a positive and reasonable range, we calculate the *partition score* $P_i$ for each

node $\Omega_i$ by a temperature-controlled softmax function:

$$P_j = \frac{exp(UCT_j/\tau)}{\sum_{j'} exp(UCT_{j'}/\tau)} \tag{3}$$

where $\tau$ is a temperature parameter controlling the smoothness of the output distribution (Hinton et al., 2015). Lower $\tau$ values make the distribution sharper and strengthen the bias towards samples from partitions with greater UCT values.

Given the partition score, the local optimizer modifies its the acquisition function value $\alpha(x; D)$ for $i$-th candidate data point $x_i$, conditioned on the history dataset $D$, as shown in Figure 2b and: $\alpha_P(x_i; D) = \alpha(x_i; D) \cdot P_{leaf_{x_i}}$, where $leaf_{\mathbf{x}}$ denotes the leaf node containing $\mathbf{x}$. With the weighted acquisition function value, the next sample $x_{next}$ to be evaluated is the one maximizing it: $x_{next} = arg\ max_{x \in X}\ \alpha_P(x; D)$. By weighing the acquisition function with the partition scores, the method prioritizes points from regions of the search space with higher sampling potential informed by the global-level navigator, while still considering the posterior estimated by the local surrogate model without constraining sampling to certain partitions. Such an augmentation of the original acquisition is aimed to make the search both globally informed and locally refined, and hence leads to a more effective exploration of the search space.

## 6 EVALUATION

In this section, we present our experiments including: 1) evaluation on synthetic benchmarks, to investigate the basic effectiveness of HiBO and compare it with related works; 2) a real-world case study of automatic DBMS configuration tuning to demonstrate its performance and practical value on high-dimensional, real-world problems; and 3) a visual analysis of the distribution of resulting partition-score-weighted acquisition function values to validate the behaviours of HiBO. The choice of real-world benchmark is discussed in Section 6.2.1.

### 6.1 SYNTHETIC BENCHMARKS

#### 6.1.1 EXPERIMENT SETUP

To evaluate HiBO on synthetic benchmarks, we compare it with several related approaches, including vanilla GP-based BO, TuRBO (Eriksson et al., 2019), LA-MCTS (Wang et al., 2020), MCTS-VS (Song et al., 2022) and CMA-ES (Lozano et al., 2006). TuRBO is selected for validating the effects of the extra space partitioning information. We compare with LA-MCTS, the current SOTA in space-partitioning-based high-dimensional BO, to validate our design's efficiency, while excluding prior methods like HOO and LA-NAS for simplicity. We also include MCTS-VS, which uses MCTS for variable selection, and CMA-ES as a baseline evolutionary algorithm. Besides, for evaluating the effectiveness of HiBO on guiding other local optimizer, we introduce HiBO-GP, where HiBO coordinates with the vanilla GP-based BO as the local optimizer.

We evaluate these algorithms on six high-dimensional synthetic benchmarks, including 200-dimensional synthetic functions Rastrigin and Levy with 200 dimensions. To evaluate how well the algorithms focus on relevant dimensions in sparse tasks, we evaluate Branin2-500D, Hartmann6-300D, Rastrigin20-200D, and Levy20-200D[1]. We run 1000 iterations for Branin2-500D and 500 iterations for the others, starting with 40 and 20 iterations of random sampling, respectively. The additional configuration details for these algorithms are shown in Appendix A.1.

#### 6.1.2 RESULTS

As observed from results on all these benchmarks (Figure 3), HiBO consistently achieves the best result across all benchmarks. Its superior performance compared to TuRBO demonstrates the advantage of incorporating global space partitioning information into the local model's acquisition strategy. HiBO also outperforms LA-MCTS on most benchmarks, indicating the effectiveness of navigating the local optimizer with partitioning information rather than limiting the sampling scope to the most promising partition. The performance gap with other algorithms widens on sparse benchmarks and becomes greatest on Hartmann6-300D, where HiBO achieves over two magnitudes lower regrets than others. This indicates HiBO's strong capacity to identify the most promising regions

---

[1]'Benchmark(x)-(y)D' indicates $x$ effective dimensions out of $y$, with the remaining $(y - x)$ being dummy dimensions.

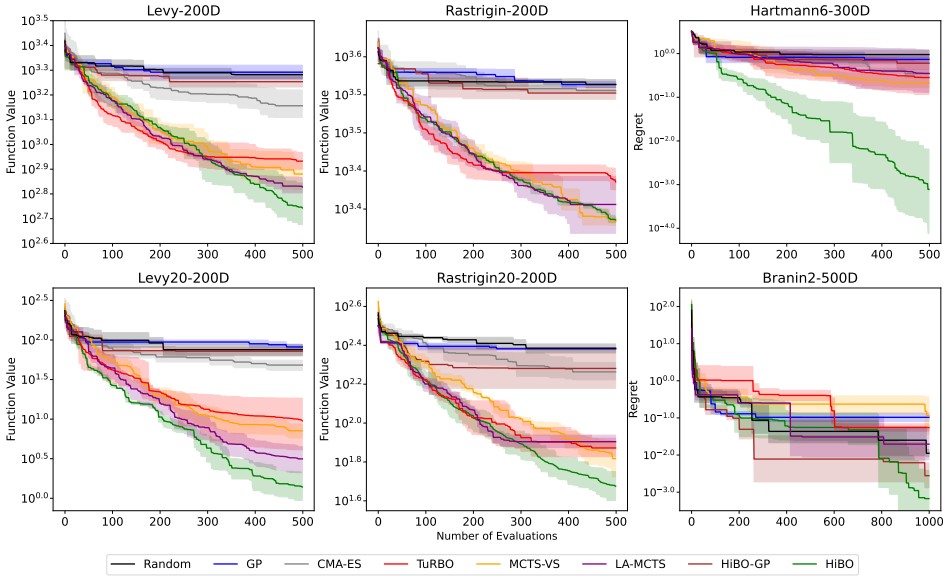

**Figure 3:** Evaluation results of algorithms on selected synthetic benchmarks.

in high-dimensional tasks. While HiBO shows similar performance to LA-MCTS on benchmarks where all dimensions are relevant, like Levy-200D, it finally finds higher-quality solutions. This suggests that HiBO may require additional iterations to adjust its partitioning strategy to cover all dimensions in dense benchmarks but can still outperform in finding optimal solutions afterwards. On the other hand, HiBO-BO is also observed to perform better than GP-BO to different extents on varying benchmarks. It can even achieve the second best and surpasses LA-MCTS in Branin2-500D task. But the improvement of HiBO-GP over GP-BO is not as stable as HiBO over TuRBO. We attribute this to differences in the local optimizer and the further guidance from HiBO can more likely produce consistent improvements over better local optimizers.

## 6.2 REAL-WORLD CASE STUDY: DBMS CONFIGURATION TUNING

### 6.2.1 EVALUATION BACKGROUND

Besides evaluation on synthetic benchmarks, we evaluate the practical value of HiBO on a more complex real-world application, specifically automatic DBMS configuration tuning. The performance of DBMS highly depends on the settings of their configuration knobs, which control various aspects of their behaviours such as memory allocation, I/O behaviors, and query optimization. However, default configurations are typically suboptimal and require further tuning for optimal performance (Zhao et al., 2023). We select DBMS configuration tuning as a critical evaluation scenario due to its significant practical importance and inherent complexities, especially for the following considerations:

- (High-dimensionality) DBMS configuration tuning involves a high-dimensional search space with over 100 configuration knobs;
- (Knob Correlation) Many of these configuration parameters are interdependent, where certain knobs are correlated and can lead to combined effects on the performance;
- (Noisy Observations) Measurements of performance data in real-world environments are much noiser compared to cases in synthetic benchmarks.

These factors make manual and heuristic tuning impractical (Huang et al., 2019; Zhao et al., 2023), making DBMS configuration tuning an ideal target for evaluating practical values of algorithms in real-world applications. In comparison, though not limited to simple mathematical synthetic functions, commonly-seen "real-world" benchmarks in literature are still large based on simulation (Jones, 2008; Wang et al., 2017; Eriksson & Jankowiak, 2021). For example, Mopta08 (Jones, 2008) originates from real-world problems in vehicle design but is abstracted into a mathematical optimization problem in practice. Considering the page limits, we included the evaluation on two

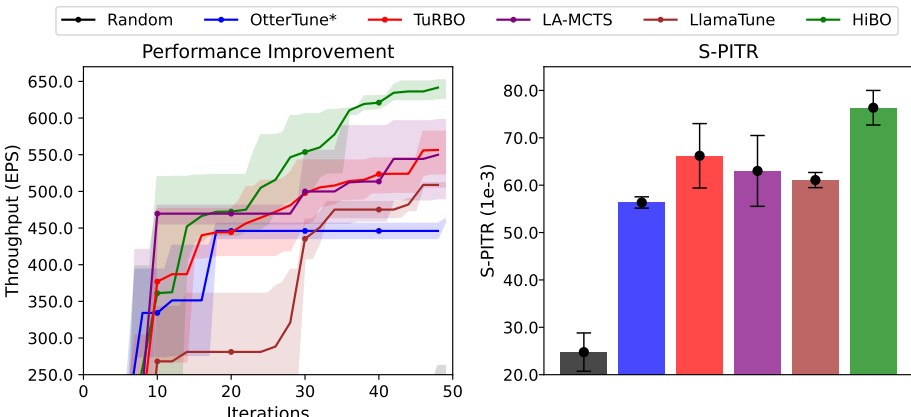

**Figure 4: Left**: Performance (throughput) evaluation of PostgreSQL being tuned by different algorithms on SysBench. Note that the figure for SysBench omits the low-performing part (throughput < 250 EPS) for readability[2]. **Right**: S-PITR measurements of experiments done with the selected algorithms, which is explained with details in Section 6.2.3.

simulated real-world benchmarks in Appendix E, including Mopta08 and Rover Trajectory Planning (Wang et al., 2017).

### 6.2.2 EXPERIMENT SETUP

Our evaluation focuses on the effectiveness of HiBO for DBMS configuration tuning without relying on extra sampling besides its own basic initial sampling. We choose PostgreSQL (version 16.3) as the target DBMS considering its accessibility and general applicability. From OnGres Inc. (2023), we manually selected 110 knobs for tuning, excluding those related to debugging, network connections, and authorization. SysBench (Kopytov, 2004) is applied as a standard OLTP workload in the experiment throughput (measured in events per second, EPS) as the target metric to be maximized. We used a limited sample budget of 50 iterations for simulating scenarios in production environments.

Selected algorithms for comparison include related BO methods and approaches specifically designed for DBMS configuration tuning: 1) Random search with uniform sampling; 2) OtterTune (Van Aken et al., 2017) without workload characterization and knob selection, noted as OtterTune$^*$; 3) TuRBO, 4) LA-MCTS, and 5) LlamaTune (Kanellis et al., 2022). The details of algorithm selection, explanation and experiment configurations can be found in Appendix A.2.

### 6.2.3 RESULTS AND ANALYSIS

Figure 4 (Left) summarizes the performance improvements of PostgreSQL on SysBench achieved by different algorithms. HiBO shows the highest throughput gains in greatest improvement rate, achieving about 28% of performance improvements over the default performance (200 EPS) than LA-MCTS and TuRBO do in 50 iterations. The comparison between HiBO and TuRBO highlights HiBO's effectiveness in guiding local models, even in complex and noisy real-world scenarios. LA-MCTS falls behind HiBO and is close to TuRBO, further indicating that simply limiting the sampling scope is insufficient, especially with limited sample budgets. Moreover, without additional pre-sampling, OtterTune* and LlamaTune also fall behind space-partitioning-based approaches.

Besides performance improvements, in real-world scenarios, tuning time and the rate of suggesting failure-causing configurations must be considered towards better practicality. A failure-causing configuration, defined as *unsafe* (Zhang et al., 2022), can degrade the performance of critical DBMS-dependent processes in the system. Therefore, we introduce the Safety-weighted Performance Improvement-Time Ratio (**S-PITR**): $S\text{-}PITR = \frac{PI}{TT+NF\cdot PE}$, where $PI$ is **p**erformance **i**mprovement, $TT$ refers to **t**uning **t**ime, $NF$ represents the **n**umber of **f**ailed (unsafe) configurations during the search and $PE$ is a the **pe**nalty value for failed configurations. This metric quantitatively measures the practicality of tuning algorithms by considering both tuning time and safety.

---

[2]which makes the curve for *Random* almost invisible.

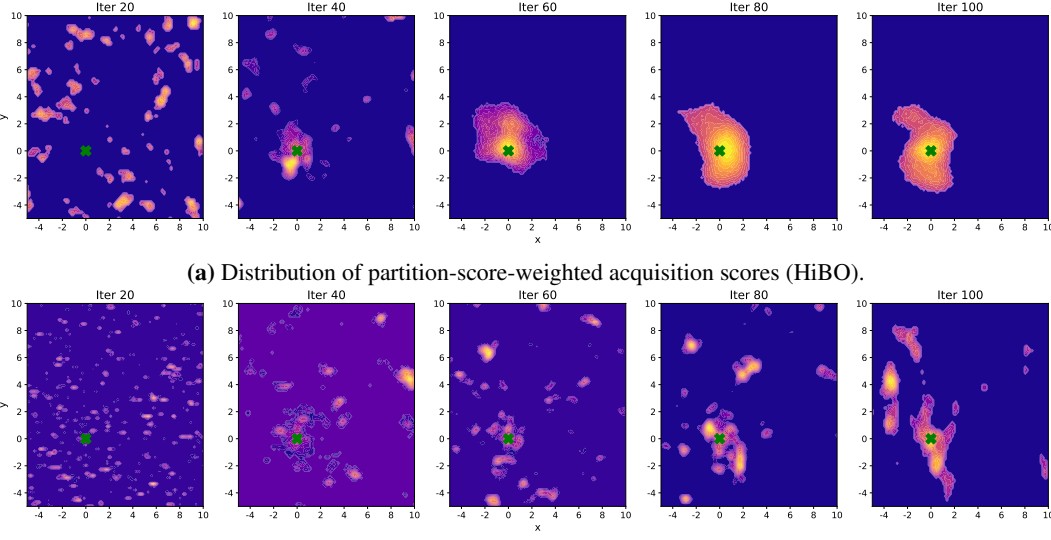

**(a)** Distribution of partition-score-weighted acquisition scores (HiBO).

**(b)** Distribution of vanilla acquisition scores (TuRBO).

**Figure 5:** Visualization of distribution of partition-score-weighted acquisition scores (HiBO) and vanilla acquisition scores (TuRBO) on the first two dimensions of Ackley2-200D across iterations. Lighter color indicates greater acquisition values and the green 'X' represents the optimal point (0, 0).

Figure 4 (Right) presents the S-PITR values for the selected algorithms while tuning PostgreSQL's configurations towards SysBench. Again, HiBO achieves the highest S-PITR value and outperforms LA-MCTS by over 20%, demonstrating its capability to navigate towards promising regions while accounting for both failure rate and computational cost. In comparison, LA-MCTS even falls behind TuRBO in terms of S-PITR, suggesting that its added complexity does not efficiently translate into practical benefits. Besides, HiBO presents ∼25% greater S-PITR scores than OtterTune* and LlamaTune. These findings highlight HiBO's adequate ability to strike a balance between performance improvement, safety, and efficiency in complex real-world DBMS tuning scenarios.

### 6.3 Visual Analysis

We visualize the acquisition function values generated by HiBO and TuRBO during optimization of the Ackley2-200D synthetic function over 100 iterations. Starting with 20 initial samples, we plot the distribution of acquisition values on the two effective dimensions as the optimization progresses. For clarity, the top 1000 acquisition values from 100x100 uniformly distributed samples are shown in each iteration. Figure 5 illustrates these results. The comparison shows that HiBO, with the extra information of space partitioning, more rapidly converges its acquisition distribution towards the optimal point, as indicated by the brightest areas near the optima. In contrast, while also moving toward the optimal region, the convergence speed of TuRBO is way slower and exhibits a less concentrated and noisier final distribution, with significant acquisition values still dispersed far from the optima. This visual analysis further validates the effectiveness of HiBO by integrating space partitioning into the acquisition strategy of the local optimizer.

## 7 Conclusion

In this paper, we introduce HiBO, a novel Bayesian Optimization algorithm designed to efficiently handle high-dimensional search spaces via a hierarchical integration of global-level space partitioning and local modeling. HiBO dynamically refines global partitioning information and seamlessly integrates it into local acquisition strategies, thereby accelerating the convergence towards global optima. Our comprehensive evaluation demonstrate its effectiveness and practical value on both synthetic benchmarks and, more importantly, on real-world systems. Additionally, the visual analysis presents a straightforward validation of its effects in navigating on complex optimization landscapes. More discussions on limitations and scalability of HiBO can be found in Appendix F and G respectively.

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

# A EXPERIMENT SETTINGS

## A.1 ALGORITHM CONFIGURATIONS FOR SYNTHETIC BENCHMARKS

- **Random Search**: Uniform sampling using function 'np.random.uniform'[3]

- **GP-BO**: Implemented using 'GaussianProcessRegressor' from scikit-learn library (Pedregosa et al., 2011), configured with a Matérn kernel ($\nu$=2.5) (Matérn, 2013) scaled by a constant factor of 1.0, and the noise level is set to 0.1, which translates to an alpha value of 0.01. The acquisition function is set to Expected Improvement (Jones et al., 1998);

- **CMA-ES**: We used the out-of-box PyCMA (Hansen et al., 2019) implementation of CMA-ES, where initial standard deviation is set to 0.5;

- **TuRBO**: hyperparameter settings are divided into two parts: 1) about the GP model: implemented using BoTorch (Balandat et al., 2020) with noise constraint interval of $[1 \times 10^{-8}, 1 \times 10^{-3}]$, a scaled Matérn kernel ($\nu = 2.5$) and Automatic Relevance Determination (ARD; Neal (2012)) lengthscales for each input dimension constrained within $[0.005, 4.0]$; 2) TuRBO-related: thresholds of consecutive successes and failures to trigger trust region size changing are set to 3 and 5 respectively, with the minimum trust region length being 0.03125 within normalized range. The acquisition function is set to Thompson Sampling (TS; Thompson (1933).

- **LA-MCTS**: Use TuRBO as the local model with the same configuration mentioned above for a separate TuRBO. The leaf size (split threshold) is set to 10 samples. The kernel type for SVM used in its MCTS is radial basis function (RBF; Vert et al. (2004)

- **HiBO**: Temperature for Softmax is set to 0.1. The limit of maximum tree depth is 5. $C_p$ is set to 0.5. Thresholds of consecutive successes and failures to trigger trust region size changing are set to 5 and 3 respectively. Use TuRBO as the local model with the same configuration mentioned above for a separate TuRBO. Use the same SVM classifier for search-tree-based space partitioning in the global-level navigator.

## A.2 EXPERIMENT SETUP FOR DBMS CONFIGURATION TUNING

The algorithms used in both synthetic and DBMS configuration tuning have the same configurations as they have in synthetic benchmark evaluation. The following are configurations for the two algorithms specifically designed for DBMS configuration tuning.

- **OtterTune***: As mentioned in Section 6.2.2, this is a simplified version of OtterTune (Van Aken et al., 2017) without workload mapping and statistical knob selection modules. It applies the vanilla GP-based BO for tuning in our setting to fit the scenario with very limited sample budget and high dimensionalities. The configurations for GP follow those used in synthetic benchmarks.

- **LlamaTune**: LlamaTune (Kanellis et al., 2022) is the most relevant work to ours in the context of DBMS configuration tuning. It directly tunes the DBMS over the high-dimensional configuration space with limited prior knowledge, employing subspace-embedding methods (Nayebi et al., 2019; Wang et al., 2016) to scale vanilla BO to high-dimensional tasks. We follow the experimental setup specified in Section 6.1 of their paper, including the use of HeSBO (Nayebi et al., 2019) random projections with 16 dimensions, a 20% biasing towards special values, and bucketizing the search space by limiting each dimension to 10,000 unique values. The base surrogate model is SMAC, implemented using the SMAC3 library (Lindauer et al., 2022).

The experiments for DBMS configuration tuning were conducted on two VM instances with identical configurations. Each of them is equipped with 8 single-core Intel(R) Xeon(R) Gold 6142 CPU @ 2.60GHz, 64GB RAM and 256GB SSD. One instance is used for executing tuning management and optimization algorithm logic. The DBMS and stored databases are stored in the other instance. One instance leverages SSH to control the other for executing DBMS workloads.

We select PostgreSQL v16.3 (PostgreSQL Global Development Group, 2023) considering its accessibility and generality. PostgreSQL is a powerful, open-source Relational DBMS that has become one of the most popular choices for managing enterprise-level databases (Merit Data Tech, 2023; Percona, 2023) given its reliability and extensibility. Importantly, PostgreSQL exposes up to more than 200 configuration knobs as shown in PGCONF (OnGres Inc., 2023) which makes it an ideal

---

[3]https://numpy.org/doc/stable/reference/random/generated/numpy.random.uniform.html

instance for our DBMS configuration tuning experiments. We collected information of all configuration knobs of PostgreSQL in PostgreSQL Co.NF (OnGres Inc., 2023). A total of **110** knobs were manually selected, excluding those related to debugging, network connections, and authorization, as they could interfere with the interaction between the DBMS controller module and the DBMS instance and have negligible impact on DBMS performance.

To simulate the limited sample budget typical in production environments, we use only 50 iterations for tuning, with the initial 10 iterations reserved for random sampling. This is also based on the observation that tuning for more iterations brings no significant improvement while resulting in extra tuning time overhead. Each iteration involves a 100-second workload execution. Configurations that fail to launch PostgreSQL or cause execution errors during execution are penalized with a throughput value of zero.

### A.3 REASONING: ALGORITHM SELECTION IN DBMS CONFIGURATION TUNING EXPERIMENTS

This section provides the clarification of our selection on BO-based approaches specifically for DMBS configuration tuning. Many BO-based methods in this field (Zhang et al., 2021; 2022; Cereda et al., 2021) and the full-version OtterTune were not included in our evaluation due to the inapplicability of their special data and different focus:

- Some methods rely on pre-collected data or additional knowledge before actual tuning as mentioned in Section 1. For example, OtterTune requires a large number of random samples to be collected, while ResTune (Zhang et al., 2021) applies workload features and observations from 34 prior tuning tasks. These requirements add substantial overhead and are impractical for direct comparison, and reproducing their setups is infeasible without the required data. Our experiments show that HiBO achieves significant throughput improvements on PostgreSQL without pre-collected data and within a limited sample budget, providing an implicit comparison of its effectiveness even without directly implementing these methods.

- Moreover, these works focus on low-dimensional tasks rather than directly optimizing high-dimensional configuration spaces. OtterTune selects important knobs for dimensionality reduction through pre-tuning sampling, while ResTune, OnlineTune (Zhang et al., 2022) and CGP-Tuner (Cereda et al., 2021) limit their evaluations to using fewer than 50 knobs through manual selection. In contrast, HiBO directly tunes the high-dimensional configuration space without requiring dimensionality reduction, prior knowledge, or complex models, while maintaining high sample efficiency.

Therefore, we only introduced the partial version of OtterTune as a baseline (OtterTune$^*$) and LlamaTune, which also applies high-dimensional BO methods.

### A.4 SYSBENCH STATEMENT TYPES

We used the script 'oltp_read_write.lua' released by the authors of SysBench (Kopytov, 2004) for benchmarking, which contains the following types of queries:

- Point selection: `"SELECT c FROM sbtest%u WHERE id=?"`

- Range selection: `"SELECT c FROM sbtest%u WHERE id BETWEEN ? AND ?"`

- Range selection with sum: `"SELECT SUM(k) FROM sbtest%u WHERE id BETWEEN ? AND ?"`

- Range selection with ordering: `"SELECT c FROM sbtest%u WHERE id BETWEEN ? AND ? ORDER BY c"`

- Range selection with distinction: `"SELECT DISTINCT c FROM sbtest%u WHERE id BETWEEN ? AND ? ORDER BY c"`

- Index update: `"UPDATE sbtest%u SET k=k+1 WHERE id=?"`

- Non-index update: `"UPDATE sbtest%u SET c=?  WHERE id=?"`

- Deletion: `"DELETE FROM sbtest%u WHERE id=?"`

- Insertion:   `"INSERT INTO sbtest%u (id, k, c, pad) VALUES (?, ?, ?, ?)"`

| Hyperparameter | Description | Range |
|---|---|---|
| $C_p$ (UCT Constant) | Balances exploration and exploitation in the UCT formula. | [0.5, 5.0] |
| $\tau$ (Softmax Temperature) | Controls the impact of global-level navigator's guidance by adjusting the smoothness of sampling potential distribution. | [0.01, 2.0] |
| Maximum Tree Depth Limit | Threshold of maximum tree depth, beyond which a restart should happen after consecutive failures. | [3, 5] |

**Table 2:** Hyperparameters in HiBO.

where "?" here stands for parameters to be automatically specified by SysBench when these queries are executed.

## B    HYPERPARAMETERS

This section summarizes the hyperparameters used in our work, including their purposes and the ranges employed during grid search to determine the optimal combination. The specific setting of algorithms used in experiments can be found in Appendix A.1 and Appendix A.2, while the ranges presented here are those we used for grid search to find the best parameter value combination:

- $C_p$: Controls the balance between exploration and exploitation in the UCT calculation (Equation 2). Larger values increase the weight of exploration. Range: [0.5, 5.0].

- $\tau$: Softmax temperature for normalizing partition scores into a positive and normalized range. Detailed interpretation and ablation studies about this hyperparameter are available in Appendix D. Range: [0.01, 2].

- **Maximum Tree Depth Limit**: Sets the limit of maximum allowable tree depth before the search process restarts after multiple times of repeated failures. It controls the tolerance of consecutive failures in one search. Range: [3, 5].

A naive grid search was used to identify the optimal combination of these hyperparameters within the specified ranges. Table 2 provides a concise summary for reference.

## C    ABLATION STUDY: ADAPTIVE MAXIMUM TREE DEPTH

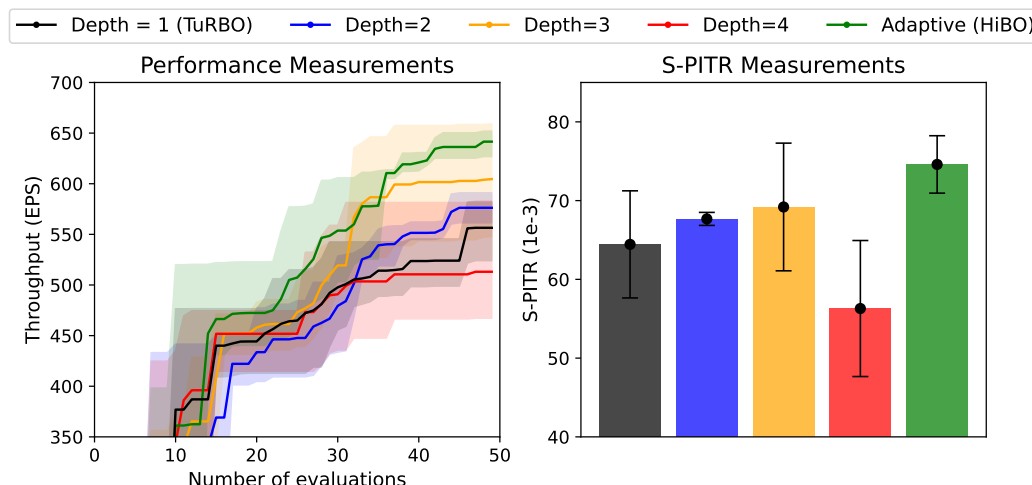

**Figure 6:** Performance improvement evaluation and measurements of S-PITR for HiBO with different maximum tree depth settings on SysBench.

In this section, we conduct an ablation study on the adaptively adjusted maximum depth of the search tree to validate its effects. As discussed in Section 4.3, we designed extra rules to dynamically adjust the maximum tree depth for the search tree depending on consecutive successes and failures, aiming

|  | Optim Exec (%) | Config Load (%) | Workload Exec (%) | S-PITR ($1e^{-3}$) |
|---|---|---|---|---|
| **Depth=1 (TuRBO)** | 2.95 ± 1.07 | 29.16 ± 5.21 | 67.89 ± 4.87 | 64.44 ± 6.80 |
| **Depth=2** | 3.98 ± 0.41 | 28.84 ± 3.40 | 67.18 ± 3.19 | 67.68 ± 0.83 |
| **Depth=3** | 4.21 ± 0.36 | 32.44 ± 1.22 | 63.35 ± 1.22 | 69.19 ± 8.11 |
| **Depth=4** | 4.75 ± 1.04 | 32.24 ± 2.99 | 63.01 ± 3.01 | 56.30 ± 8.64 |
| **Adaptive (HiBO)** | **4.31 ± 0.91** | 29.19 ± 3.28 | 66.50 ± 2.98 | **74.59 ± 3.64** |

**Table 3:** Iteration time breakdown during applying HiBO for DBMS configuration tuning under different maximum tree depth settings. 'Optim Exec' refers to the proportion of time spent on optimization algorithm execution. 'Config Load' and 'Workload Exec' similarly indicate the time proportion for loading the suggested configuration on the DBMS and executing the benchmark workload.

to 1) reduce unnecessary computational cost and 2) balance exploitation and exploration by adjusting the range covered by resulting search space partitions. To assess the effects of these designs, we evaluate HiBO under different tree depth settings, considering both efficiency and computational cost.

For this experiment, we use DBMS configuration tuning with workload being SysBench as the benchmark, following the same configuration setup described in Section 6.2.2. This setup is expected to provide more practical insights into the computational cost reduction in real-world workloads than using synthetic functions. For comparing with proposed dynamic tree depth setting, the maximum depth of the search tree is fixed at constant values of 1, 2, 3, and 4. It is important to note that the depth being 1 represents a special case where the search tree consists of a single root node, and the algorithm degrades to plain TuRBO because no partitions are split for guiding the local search. Performance data from previous experiments using dynamic adjustment are used, denoted by 'Adaptive (HiBO)'.

In addition to performance measurements, considering the maximum tree depth is closely related to the tuning time cost, S-PITR is calculated to evaluate performance improvement relative to tuning time and safety factors for a more comprehensive evaluation. The results are presented in Figure 6, from which we have the following findings:

- Increasing tree depth from 1 to 3 in constant-depth search trees improves both performance and S-PITR, indicating the effectiveness of HiBO's partition-level guidance. The growing S-PITR shows that the additional computational cost of increasing tree depth is outweighed by the performance gains.

- Performance declines when the depth increases to 4, with results falling below those of plain TuRBO. This suggests that a tree depth of 4 introduces excessive bias toward limited space partitions, leading to over-exploitation. Additionally, the higher computational cost of constructing a 4-layer tree further reduces the S-PITR score.

- Our proposed dynamic tree-depth adjustment achieved the highest performance improvements, with superior S-PITR scores. This demonstrates that our approach effectively balances exploitation and exploration with reasonable tuning time and failure rates.

Besides S-PITR, we provide a detailed breakdown of the iteration time on average during DBMS configuration tuning, which comprises the time for configuration loading, workload execution, and optimization execution. Table 3 summarizes the proportions for HiBO with different maximum tree depth settings. From the table, we observe that while HiBO introduces additional computational costs during the optimization phase compared to TuRBO, the increase remains minimal. Even with a fixed tree depth of 4, the added proportion of execution time is within 2% on average per iteration. The adaptive tree depth control maintains computational costs between those of fixed depths of 3 and 4 while delivering the best performance improvements and highest S-PITR scores.

## D ABLATION STUDY: TEMPERATURE

This section presents an ablation study on the effect of the hyperparameter $\tau$, which serves as the temperature in the Softmax (3). $\tau$ controls the influence of global-level space partitioning by adjust-

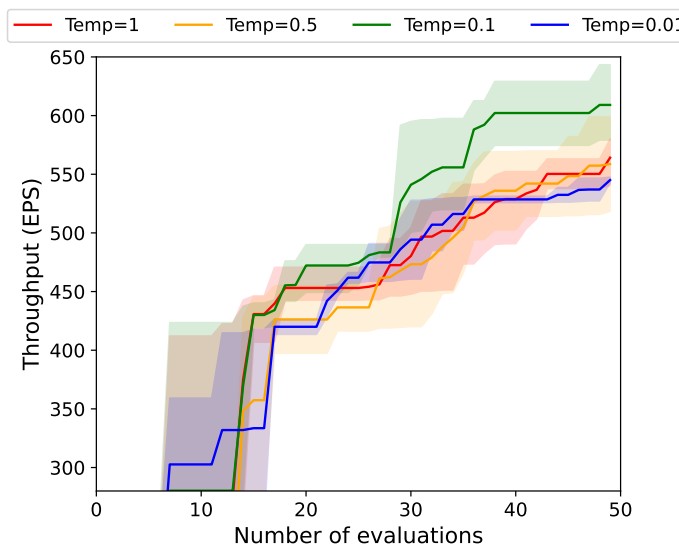

**Figure 7:** Performance (throughput) evaluation of PostgreSQL being tuned on SysBench by HiBO with different temperature settings.

ing the smoothness of the estimated distribution of sampling potential across partitions, where the two extreme cases of its value can lead to the following effects:

- **Extremely high** $\tau$: Results in an overly smoothed distribution, reducing the impact of global-level partitioning and diminishing differences between partitions;

- **Extremely low** $\tau$: Results in a peaky distribution and assigns overwhelming weight to the partition with the highest sampling potential, amplifying the navigator's influence.

We evaluated HiBO's performance on DBMS configuration tuning with $\tau$ values of 1, 0.5, 0.1, and 0.01. Other experiment settings follow those described in Section 6.2.2 and Appendix A.

The evaluation results are presented in Figure 7. The results show that when temperature is set to 0.1, the performance improvement and improvement rate is evidently better than other temperature settings. While the other three temperatures have similar tuning effects with setting temperature to 0.01 resulting in slight degradation on final performance improvements. These findings suggest the existence of an optimal range for $\tau$. For tuning configurations of PostgreSQL towards workload SysBench, $\tau = 0.1$ strikes a good balance between these two ends of phenomenon and achieved better DBMS performance improvements on SysBench.

# E  EVALUATION ON ADDITIONAL REAL-WORLD BENCHMARKS

Though DBMS configuration tuning has been introduced as a unique challenging benchmark for evaluating BO-based algorithms on high-dimensional search space, we still evaluated HiBO on two widely recognized real-world benchmarks in prior works (Eriksson et al., 2019; Eriksson & Jankowiak, 2021; Shen & Kingsford, 2023; Rashidi et al., 2024) to further validate its performance, which include the following two benchmarks:

- Mopta08 (124D; Jones (2008)): This task involves minimizing the mass of a vehicle by optimizing 124 design variables that control materials, gauges, and vehicle shape. The optimization is subject to 68 constraints, including performance, safety, and feasibility requirements. Mopta08 is a high-dimensional, constrained optimization problem widely used to benchmark BO methods in real-world-inspired engineering design scenarios.

- Rover Trajectory Planning (60D; Wang et al. (2017)): In this task, the objective is to maximize the total reward collected by a rover along a planned trajectory. The trajectory is defined by optimizing the coordinates of 30 points in a 2D plane, resulting in a 60-dimensional search

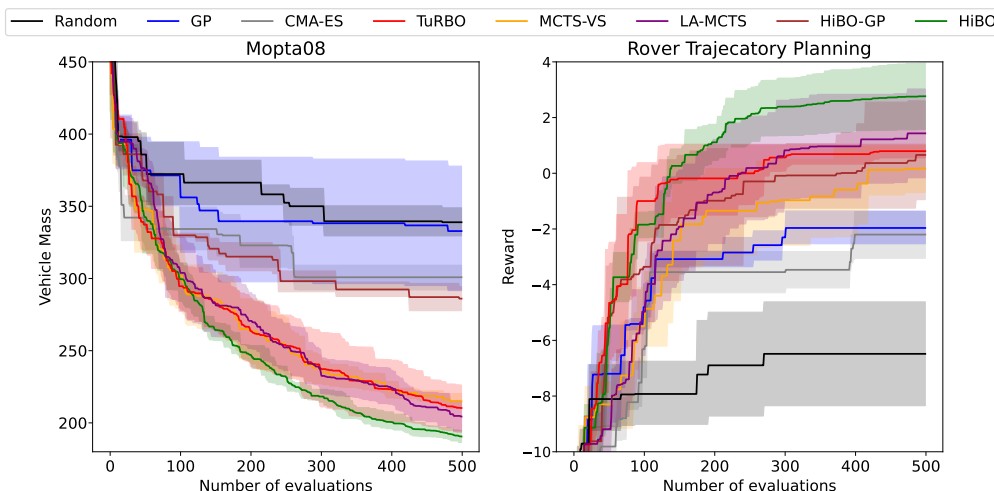

**Figure 8:** Evaluation results of algorithms on Mopta08 (Left) and Rover Trajectory Planning (Right).

space. This benchmark reflects practical challenges in robotics, particularly in path planning and navigation under constraints.

The results for these additional benchmarks are presented in Figure 8. As shown in Figure 8, HiBO demonstrates competitive performance across both tasks, achieving an 20 more units of vehicle mass on Mopta08 and an 1 more reward value on Rover Trajectory Planning than LA-MCTS on average. Furthermore, HiBO-GP consistently outperforms standard GP-BO in these benchmarks. Specifically, HiBO-GP achieves over 25% less vehicle mass on Mopta08 and approximately 2 additional reward values on Rover Trajectory Planning compared to vanilla GP-BO. These results further underscore HiBO's effectiveness and its ability to tackle complex, high-dimensional optimization problems frequently encountered in practical applications.

## F LIMITATIONS

We summarized the following limitations with HiBO based on reasoning and experiment results:

- **Performance Improvment on Dense Benchmarks**. One limitation of HiBO lies in its relatively less pronounced improvement over the local optimizer on dense high-dimensional benchmarks compared to sparse benchmarks. While HiBO demonstrates significant performance gains on sparse benchmarks, its improvement is less evident on dense tasks where all dimensions are uniformly effective such as Levy-200D and Rastrigin-200D). Our interpretation is that in sparse spaces, HiBO efficiently identifies promising regions to guide the local optimizer and it is one of its greatest advantages. While in dense spaces, it requires more samples to achieve accurate estimations across the entire space than in sparse scenarios. But HiBO still converges efficiently towards the optimal point in these tasks and achieves the best in Levy-200D and Rastrigin-200D.

- **Unobservable Configurations**. HiBO may also struggle in scenarios where certain critical configuration variables are hidden or unobservable during optimization. Such situations can arise in practical tasks where parts of the configuration space are inaccessible due to hardware constraints, legacy systems, or incomplete problem specifications. In these cases, the global-level navigator may build an inaccurate model of the search space, leading to suboptimal partitioning and sampling.

## G SCALABILITY

### G.1 SCALABILITY WITH RESPECT TO INCREASING DIMENSIONALITY

This section discusses the scalability of HiBO with respect to dimensionality. Given that the primary modification to the original local optimizer lies in the global-level navigator, our analysis focuses on

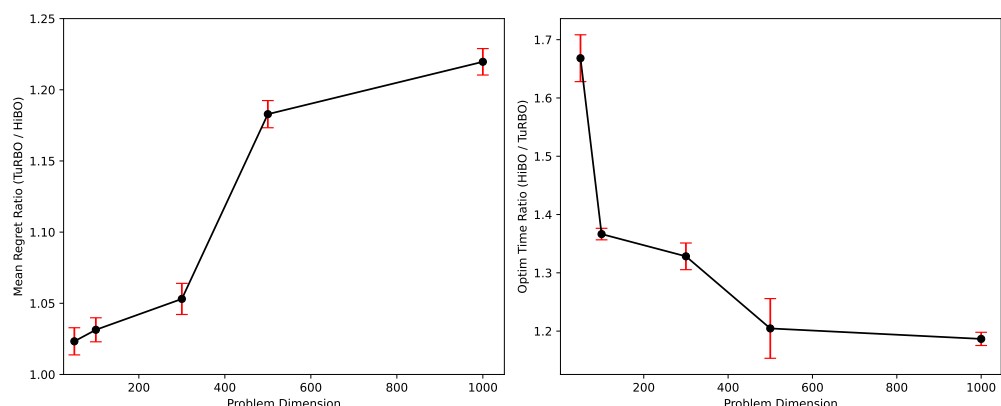

**Figure 9:** Comparison of HiBO and TuRBO across varying dimensionalities on the synthetic Levy benchmark. (**Left**) The ratio of mean regret (RMR) between TuRBO and HiBO ($\frac{r\bar{e}gret_{TuRBO}}{regret_{HiBO}}$), indicating HiBO's performance advantage over TuRBO increases with dimensionality. (**Right**) The ratio of optimization time (ROT) for HiBO relative to TuRBO ($\frac{Time_{HiBO}}{Time_{TuRBO}}$), showing a decreasing trend as dimensionality increases, reflecting the reduced relative overhead of HiBO's global-level navigator.

comparing the performance and timing costs of HiBO and TuRBO on the synthetic Levy benchmark across varying dimensions, as shown in Figure 9. The algorithm settings follow those detailed in Appendix A, and the number of iterations is set to 500. Figure 9 (Left) depicts the ratio of mean regret (RMR) between TuRBO and HiBO ($\frac{r\bar{e}gret_{TuRBO}}{regret_{HiBO}}$), while Figure 9 (Right) illustrates the ratio of optimization time (ROT) for HiBO relative to TuRBO ($\frac{Time_{HiBO}}{Time_{TuRBO}}$).

The results indicate that RMR increases steadily with dimensionality, suggesting that TuRBO's regret grows more significantly than HiBO's as dimensionality rises. This demonstrates that HiBO's improvement over plain TuRBO scales effectively with increasing problem dimensions. On the other hand, although ROT remains greater than 1, reflecting the additional time cost brought by HiBO, ROT decreases as dimensionality increases. This trend suggests that as the dimensionality grows, the local optimizer (i.e. TuRBO) experiences a heavier computational burden and suffers from poorer scalability compared to the global-level navigator. As a result, the relative overhead introduced by the global navigator becomes less significant as the local optimizer becomes the dominant computational component in increased dimensionalities

In conclusion, HiBO exhibits robust scalability with increasing dimensionality, as evidenced by its growing performance advantage over the local optimizer and the decreasing relative computational overhead of its global-level navigator.

### G.2 SCALABILITY WITH RESPECT TO LARGE-SCALE DATASETS

Besides the scalability with respect to dimensionality, here we provide clarification on the scalability with respect to large-scale data samples. In fact, there is no easy answer to such scalability of HiBO especially because the search tree depth is adaptively controlled and the total computational cost can vary from different benchmarks. Further, it also depends on the types and specified configurations of clustering and classification algorithms used for search space partitioning, which can have different computational complexity. For example, the classifier used in the search space partitioning can be SVMs based on linear kernels with $O(n)$ complexity or non-linear kernels with $O(n^2)$ or $O(n^3)$, where $n$ is the number of data samples. However, the focus of this paper is the design of the hierarchical framework and we do not conduct specific analysis on designing specific component combination to large scale dataset.

Moreover, HiBO is designed for high-dimensional tasks with limited sample budgets, which aligns with real-world scenarios where objective evaluations are expensive to be evaluated. The DBMS configuration tuning serves as a typical example because ideally we expect an important DBMS instance can keep running instead of spending time on tuning. Correspondingly, related works in the

field of DBMS configuration tuning typically control their sample budget to less than 500 iterations (Zhang et al., 2021; 2022; Cereda et al., 2021; Kanellis et al., 2022). This work even adopts the very limited sample budget (50 iterations) to form a challenging benchmark, while HiBO can improve the default performance by more than 2 times within this sample budget. In fact, tuning towards opaque problems within limited sample budget is the one of core design principles of BO and HiBO adequately exemplifies this consideration. From this perspective, scalability with respect to large scale dataset is out of scope of this work.

