# OpenReview forum: "HiBO: Hierarchical Bayesian Optimization via Adaptive Search Space Partitioning"
_ICLR.cc/2025/Conference — Submitted to ICLR 2025_

### Official Review · Reviewer_D9Dm · 2024-10-31

**Soundness:** 2
**Presentation:** 3
**Contribution:** 2
**Rating:** 6
**Confidence:** 3

**Summary:**

In the paper, the authors propose HiBo which uses a hierarchical search space partitioning algorithm to reduce the search space and then apply local BO-based optimizer to sample from the most critical regions. In addition, HiBo uses the upper confidence bounds to evaluate tree node’s sampling potential and hence balance to explorations and exploitations,

**Strengths:**

The paper studies an important problem of optimizing a black-box function in high dimensional space.

The overall concept is sound and intuitive, which serves as both a strength and a limitation, as many of the observations rely on heuristics amd emprical observations.

The idea of determining tree depth based on consecutive successes or failures is sound and interesting.

**Weaknesses:**

Why does HiBo reply on a binary serach tree instead of paritioning each level to multiple clusters? (like R tree) Also doe the serach tree need to be a balanced tree?

I find it challenging to fully appreciate Rule 1 and Rule 3. I think these rules make sense but the author should perhaps give better insights on these two rules.

In fact, one may argue that DBMS tunning is suitable for BO becuase the number of important knobs can be small?

Perhaps I missed it, but what is the workload in sysbench (read-only, write-only, mixed)?

**Questions:**

See Weakness.

---

> ### Author Response · Authors · 2024-11-22
> **Response to Reviewer D9Dm (W1, W2, W3)**
>
> ## W1
> > W1. Why does HiBO reply on a binary serach tree instead of paritioning each level to multiple clusters? (like R tree) Also does the serach tree need to be a balanced tree?
>
> Thanks for pointing out the design choice of binary search tree. Indeed this can be a valid direction of extending the current work but the effectiveness of our framework does **not** rely on the number of sub-partitions generated by each partitioning operation.
>
> The partitioning is aimed to differentiate the sampling potential of the generated sub-partitions and provides information of distribution of sampling potential over the search space. This should also be applied to general K-ary cases only if the difference of sampling potential among sub-partitions can be effectively maximized by the clustering- and classification-based partitioning operations. The binary search tree was chosen in our current design for simplicity and computational efficiency, as it minimizes the overhead of managing and evaluating a larger number of partitions at each level.
>
> Regarding the need for a balanced tree, HiBO's framework does not strictly require the tree to be perfectly balanced. Empirically,  a threshold on the quantitative metric of the clustering quality to determine whether this partitioning operation can even be set to avoid low-quality splitting and hence result in unbalanced trees. But the core, again, of building this tree is to differentiate the sampling potential of sub-partitions generated by recursive partitioning, whether the tree balanced or not is not the focus of our work as we care more about whether the generated distribution of sampling potential over partitions is informative and effective.
>
> ## W2
> > W2. I find it challenging to fully appreciate Rule 1 and Rule 3. I think these rules make sense but the author should perhaps give better insights on these two rules.
>
> Sorry for any confusion resulted from the rules in that part and we are happy to provide some clarification here::
>
> 1. **Rule 1:** Rule 1 is more like an implementation tip for conveniently tracking the depth of the search tree by breadth-first search (BFS). In our algorithm, the tree height corresponds to the number of bifurcations required to form the partitions represented by the leaf nodes. Greater tree depth results in smaller partitions, emphasizing more specific regions for exploitation. For balancing the exploitation and exploration, we implement the logic based on BFS to track the depth when building the tree and stop expanding the tree when the depth reaches the maximum depth.
> 2. **Rule 3:** We sincerely apologize for the typo in Rule 3 and thank you very much for pointing it out. The phrase “falls below a threshold” should instead read “**goes beyond a threshold**.” When the tree depth goes beyond the threshold, it indicates it has encountered multiple times of consecutive failures, the region covered by the partition with the greatest sampling potential becomes limited and the search is suspected to get stuck into a non-promising region. This is similar to the rule applied by TuRBO which restarts the search when trust region size becomes too small after multiple times of consecutive failures. **We have modified this statement in the revised version of our paper.**
>
>
> ## W3
> > W3. In fact, one may argue that DBMS tunning is suitable for BO becuase the number of important knobs can be small? Perhaps I missed it, but what is the workload in sysbench (read-only, write-only, mixed)?
>
> Thanks for raising the concern on DBMS configuration tuning. The fact is basically correct because DBMS configuration tuning does have limited number of important knobs, but this does not diminish our conclusion that HiBO's effectiveness. The total dimensionality of this task is still high while the sample budget is very limited. This scenario is similar to the sparse synthetic benchmarks with a small subset of effective dimensions (e.g. Levy20-200D) though DBMS configuration tuning is way more noisy and more challenging. Identifying the potential region in a high-dimensionality space for search is non-trivial, reflected by the suboptimal performance of other algorithms in this task.
>
> To further validate HiBO, we added two additional real-world benchmarks in Appendix E, including Mopta08 (124D) and Rover Trajectory Planning (60D), which are not sparse. HiBO outperformed other methods on both benchmarks, and HiBO-GP (HiBO with GP-based BO as the local optimizer) also surpassed vanilla GP-BO. Additionally, experiments on 200-dimensional Levy and Rastrigin tasks indicate that HiBO is effective for fully high-dimensional tasks.
>
> As for the question about workload in SysBench, sorry for missing the detail of the workload. The workload in SysBench is mixed including both read and write requests and the details of the SQL statements in this workload can be found in the newly added Appendix **A.3**

---

> > ### Author Response · Authors · 2024-11-22
> > **Response to Reviewer D9Dm (Thank you)**
> >
> > Thank you very much for your insightful suggestions and feedbacks. We hope that our responses and the revisions to the paper address your concerns and clarify any ambiguities. If you feel that these responses and changes improve the quality and clarity of the work, we kindly ask you to consider raising your score. We are happy to provide more clarification if you have additional concerns or feedbacks.

---

> > > ### Comment · Reviewer_D9Dm · 2024-11-26
> > > **Thank you**
> > >
> > > Thank you for your thoughtful responses. I will keep my score.
> > >
> > > Scalability is, in my view, a key motivation for this work, and it plays a crucial role in the applications like DBMS. Compared to some of the algorithms in Table 1, it indeed seems like HiBO has better scalability. I think the author can further improve the paper to address the scalability concern in Reviewer X5cm and 6Fsx.
> > >
> > > Looking at the appendix C, I believe Table 3 oversimplifies the details, and presenting the results as percentages might not be the most effective approach. For instance, the table states that 'e HiBO introduces additional computational
> > > costs during the optimization phase compared to TuRBO, the increase remains minimal .. within 2%.' However, it also shows that the optim execution time increased by 1.5x when transitioning from depth 1 to HiBo. Table 3 should also include S-PITR scores since the claim is the adaptive tree has highest S-PITR score.

---

> > > > ### Author Response · Authors · 2024-11-28
> > > > **Response to Reviewer D9Dm (Paper Revision)**
> > > >
> > > > ## Scalability
> > > >
> > > > > Scalability is, in my view, a key motivation for this work, and it plays a crucial role in the applications like DBMS. Compared to some of the algorithms in Table 1, it indeed seems like HiBO has better scalability. I think the author can further improve the paper to address the scalability concern in Reviewer X5cm and 6Fsx.
> > > >
> > > > ### Pointers
> > > >
> > > > Thank you very much for further raising the concerns on the scalability. But we suspect there can be two different interpretations of the "scalability": the scalability with respect to increased dimensionality of the problem and the scalability with respect to the large scale dataset. Inspired by your suggestions, we additionally included an extra **Appendix G** for addressing the concerns of Scalability. We discussed the scalability of this algorithm in terms of dimensionality and data samples. As discussed in previous response provided for Reviewer X5cm and 6Fsx, the scalability w.r.t large scale dataset is not commonly discussed for BO-based algorithms towards optimization **within limited sample budget** and the clarification is put in **Appendix G.2 (line 1120 - line 1140)**.
> > > >
> > > > But we further included another part of experiments on evaluating HiBO and TuRBO on Levy benchmark with increasing dimensionalities in **Appendix G.1 (line 1077 - line 1119)**, described as following:
> > > >
> > > > ### Result Analysis
> > > >
> > > > Given that the primary modification to the original local optimizer lies in the global-level navigator, our analysis focuses on comparing **the performance and timing costs of HiBO and TuRBO on the synthetic Levy benchmark for 500 iterations across varying dimensions**, as shown in **Figure 9**. Figure 9 (**Left**) depicts the ratio of mean regret (RMR) between TuRBO and HiBO ($\overline{regret}\_{TuRBO} / \overline{regret}\_{HiBO}$), while Figure 9 (**Right**) illustrates the ratio of optimization time (ROT) for HiBO relative to TuRBO ($Time_{HiBO} / Time_{TuRBO}$). Regret is defined as the difference between the optimal objective value and the value achieved by the algorithm at a given iteration, with mean regret representing the average regret over the entire optimization process. **Lower mean regret indicates better algorithm performance.**
> > > >
> > > > Figure 9 (Left) indicates that RMR increases steadily from ~1.03 to ~1.22 with dimensionality increasing from 50 to 1000, suggesting that TuRBO's mean regret on Levy grows more significantly than HiBO's as dimensionality rising. In other words, HiBO’s performance improvement over plain TuRBO scales effectively with increasing problem dimensions.
> > > >
> > > > On the other hand, as Figure 9 (**Right**) shows, although ROT remains above 1 due to the additional time cost brought by the global-level navigator of HiBO, ROT decreases from ~1.66 to ~1.18 as dimensionality increasing. This trend suggests that as the dimensionality grows, the local optimizer (i.e. TuRBO) experiences a heavier computational burden and suffers from poorer scalability compared to the global-level navigator. As a result, the relative overhead introduced by the global navigator becomes less significant as the local optimizer becomes the dominant computational component in increased dimensionalities.
> > > >
> > > > These results empirically demonstrate HiBO’s robust scalability with increasing dimensionality, as evidenced by its growing performance advantage over the local optimizer and the decreasing relative computational overhead of its global-level navigator.
> > > >
> > > > -------------------
> > > >
> > > > ## Revised Table 3
> > > >
> > > > > Looking at the appendix C, I believe Table 3 oversimplifies the details, and presenting the results as percentages might not be the most effective approach. For instance, the table states that 'e HiBO introduces additional computational costs during the optimization phase compared to TuRBO, the increase remains minimal .. within 2%.' However, it also shows that the optim execution time increased by 1.5x when transitioning from depth 1 to HiBo. Table 3 should also include S-PITR scores since the claim is the adaptive tree has highest S-PITR score.
> > > >
> > > > Thanks for the additional concern on Table 3 of lacking S-PITR scores. In fact, just in Appendix C, we already included the bar plot of S-PITR scores for each depth setting in **Figure 6 (line 896 to line 915)**. But it is a good point to also include the accurate statistics of S-PITR scores in Table 3 and we have revised this table in the newer version (see **line 918 - line 926**).

---

### Official Review · Reviewer_6Fsx · 2024-11-01

**Soundness:** 3
**Presentation:** 2
**Contribution:** 2
**Rating:** 5
**Confidence:** 4

**Summary:**

This paper presents HIBO, a hierarchical algorithm for optimizing black-box functions in high-dimensional search spaces. HIBO enhances traditional Bayesian optimization by incorporating global search space partitioning into the local acquisition strategy. It outperforms state-of-the-art methods on synthetic benchmarks and demonstrates effectiveness in real-world applications, such as database management system configuration tuning.

**Strengths:**

S1. The core concept is straightforward and comprehensible.
S2. The topic addressed is intriguing and relevant to current industry needs.
S3. Experiments show the method outperforms existing solutions, proving its effectiveness.

**Weaknesses:**

W1. The motivation is unclear. It is not evident what the challenges are in introducing structural information of the hierarchy. What are the criteria for optimally utilizing the so-called structural information?
W2. Although this paper dedicates considerable space to discussing the integration of partitioning information, its novelty is limited. Many existing techniques are well-known, which undermines its originality.
W3. There are few real-world experiments, making it difficult to support the claims about black-box functions.

**Questions:**

Q1. The paper mentions that the HIBO algorithm outperforms existing methods in high-dimensional search spaces. Could you specify its performance improvements across different dimensions?
Q2. When introducing the HIBO algorithm, it mentions using a search tree for space partitioning. Does this approach encounter issues with excessive computational complexity when handling large-scale data? If so, how can this be addressed?
Q3. The HIBO algorithm combines global and local optimization strategies, so how is the optimal balance between the global navigator and the local optimizer determined in practical applications?

---

> ### Author Response · Authors · 2024-11-22
> **Response to Reviewer 6Fsx (W1 & W2)**
>
> ## W1 & W2
> > W1. The motivation is unclear. It is not evident what the challenges are in introducing structural information of the hierarchy. What are the criteria for optimally utilizing the so-called structural information?
> >
> > W2. Although this paper dedicates considerable space to discussing the integration of partitioning information, its novelty is limited. Many existing techniques are well-known, which undermines its originality.
>
> ### Motivation
> Thanks for sharing your concern on the motivation and sorry for any unclear expression in our paper about this. What we hope to emphasize is the **insufficiency of prior works adopting search space partition for scaling BO to high-dimensional search space**. They generate search trees with the tree's structure itself conveying the information of the estimated distribution of sampling potential over the search space, but they **only choose the only region with greatest estimated sampling potential** for confining the sampling scope into it. We think this is a waste of generated information especially considering the complexity of building this tree. On the other hand, when the initial sample budget is limited, the quality of built search tree has not been good enough yet and confining the sampling scope following the tree can significantly prevent effective exploration over the search space. Our experimental results (e.g., Rastrigin20-200D, Hartmann6-300D, and DBMS configuration tuning) demonstrate LA-MCTS, as the SOTA method of this paradigm, can only perform comparably to plain TuRBO and fall behind HiBO. And we **revised our paper to emphasize the motivation in Introduction**.
>
> -------------
>
> ### Novelty
> Instead of strictly confining the sampling scope, we allow the sampling to span across the regions and aim to **leverage the information from the partitioning process** for guiding the search in a **more flexible and comprehensible way**. The guidance provided by the global-level navigator is integrated into the local optimizer's acquisition strategy, which is general for almost all BO-based algorithms and easy to implement. To the best of our knowledge, we are **the first to combine the search space partitioning information with the acquisition strategies of BO**. Besides, the adaptive control on maximum tree depth proposed in this work is empirically validated to be an effective design to balance exploitation and exploration while reducing the computational cost.
>
> Additionally, we are the **first to introduce DBMS configuration tuning as a practical benchmark for evaluating BO algorithms' effectiveness**. This benchmark presents a high-dimensional, noisy, and correlated search space, closely resembling real-world scenarios. The inclusion of this task allows us to effectively evaluate HiBO’s sample efficiency on practical high-dimensional optimization problems. From the results presented in Figure 4, HiBO achieves about 28% greater throughput improvement over the default configuration than LA-MCTS within a very limited sample budget (50), indicating the soundness of our motivation and effectiveness of our methods.

---

> > ### Author Response · Authors · 2024-11-22
> > **Response to Reviewer 6Fsx (W3)**
> >
> > ## W3
> > > W3. There are few real-world experiments, making it difficult to support the claims about black-box functions.
> >
> > Thanks for the good point to raise the concern about the real-world experiments. We understand that in the original submission we missed experiments on some popular real-world benchmarks in Bayesian Optimization (BO) literature. However, we did introduce a more practical real-world benchmark: the DBMS-based configuration tuning task. This task involves a high-dimensional search space, correlated configurations, and noisy performance observations and was conducted in real-world DBMSs. These characteristics make it more representative of real-world challenges than simulated "real-world" tasks, providing a effective evaluation of BO-based algorithm's practical values.
> >
> > Though DBMS configuration tuning has been introduced as a unique challenging benchmark for evaluating BO-based algorithms on high-dimensional search space, we still evaluated HiBO on two widely recognized real-world benchmarks in prior works to further validate its performance, which include the following two benchmarks:
> >
> > - Mopta08 (124D; Jones 2008): This task involves minimizing the mass of a vehicle by optimizing 124 design variables that control materials, gauges, and vehicle shape. The optimization is subject to 68 constraints, including performance, safety, and feasibility requirements. Mopta08 is a high-dimensional, constrained optimization problem widely used to benchmark BO methods in real-world-inspired engineering design scenarios.
> > - Rover Trajectory Planning (60D; Wang et al. 2017): In this task, the objective is to maximize the total reward collected by a rover along a planned trajectory. The trajectory is defined by optimizing the coordinates of 30 points in a 2D plane, resulting in a 60-dimensional search space. This benchmark reflects practical challenges in robotics, particularly in path planning and navigation under constraints.
> >
> > We added this additional part of evaluation about the real-world experiments in **Appendix E of the revised paper**. As shown in **Figure 8**, HiBO demonstrates competitive performance across both tasks, achieving an 20 more units of vehicle mass on Mopta08 and an 1 more reward value on Rover Trajectory Planning than LA-MCTS on average. Furthermore, HiBO-GP consistently outperforms standard GP-BO in these benchmarks. Specifically, HiBO-GP achieves over 25% less vehicle mass on Mopta08 and approximately 2 additional reward values on Rover Trajectory Planning compared to vanilla GP-BO. These results further underscore HiBO’s effectiveness and its ability to tackle complex, high-dimensional optimization problems frequently encountered in practical applications.
> >
> > References:
> >
> > - Jones 2008. Large-scale multi-disciplinary mass optimization in the auto industry
> > - Wang et al. 2017 Batched high-dimensional Bayesian Optimization via structural kernel learning

---

> > > ### Author Response · Authors · 2024-11-22
> > > **Response to Reviewer 6Fsx (Q1 & Q3)**
> > >
> > > ## Q1
> > > > Q1. The paper mentions that the HIBO algorithm outperforms existing methods in high-dimensional search spaces. Could you specify its performance improvements across different dimensions?
> > >
> > > Thanks for the question about performance improvements over different dimensions. In fact, we designed our evaluation experiments with considering this need, and HiBO’s performance was evaluated on benchmarks spanning diverse dimensionalities. Specifically, the experiments included:
> > >
> > > 1) synthetic benchmarks: 200D (Levy, Rastrigin and their corresponding sparse versions), 300D (Hartmann6) and 500D (Branin) synthetic tasks;
> > > 2) a 110D real-world DBMS configuration tuning problem,
> > > 3) 2 additional real-world benchmarks: the 124D Mopta08 constrained vehicle design task, and the 60D Rover Trajectory Planning task (See **Appendix E** in the revised version)
> > >
> > > HiBO consistently outperformed existing methods over these benchmarks. Besides simply the performance improvement measurements, it also exhibits adequate practical value with respect to its tuning time cost and failure rate in the real-world DBMS configuration task. These results highlight HiBO’s scalability and adaptability across varying dimensions and problem types.
> > >
> > >
> > > ## Q3
> > > > Q3. The HIBO algorithm combines global and local optimization strategies, so how is the optimal balance between the global navigator and the local optimizer determined in practical applications?
> > >
> > > It is a good point to ask about balancing between global navigator and local optimizer. The global navigator essentially provides weights for the local optimizer's acquisition function based on its estimated distribution of sampling potential over the space partitions. So our idea of balancing the amount of guidance from the global navigator to the local optimizer is to control the smoothness of the distribution of sampling potential by the hyperparameter $\tau$​ in the softmax.  The two extreme cases of $\tau$'s value can lead to the following effects:
> > >
> > > - **Extremely high $\tau$:** Results in an overly smoothed distribution, where the difference between biases assigned to different space partitions becomes negligible. Eliminates the impact of global-level partitioning and diminishing differences between partitions;
> > > - **Extremely low $\tau$**: Results in a peaky distribution and assigns overwhelming weight to the partition with the highest sampling potential, amplifying the navigator's influence.
> > >
> > > In fact, we did an additional ablation study on this hyperparameter by evaluating HiBO on DBMS configuration tuning task with different temperature, and the experiment does indicate there can exist some optimal regions for balancing the two components. Please see **Appendix D in the revised paper** for more details about this part of experiments.
> > >
> > > For the main experiments in the Evaluation part, we applied naive grid search to determine the optimal parameter combination involving the temperature and set this temperature to be 0.1 for striking a good balance between the two components and achieving good empirical results on the benchmarks.

---

> > > > ### Author Response · Authors · 2024-11-22
> > > > **Response to Reviewer 6Fsx (Q2)**
> > > >
> > > > ## Q2
> > > > > Q2. When introducing the HIBO algorithm, it mentions using a search tree for space partitioning. Does this approach encounter issues with excessive computational complexity when handling **large-scale data**? If so, how can this be addressed?
> > > >
> > > > Thanks for raising the concern about the scalability of our algorithm to large scale dataset. During our experiments we **did not meet any issues resulting from the extra computational complexity**. Though HiBO introduces the extra global-level component, it also incorporates the Adaptive Maximum Tree Depth mechanism to balance the exploitation and exploration while preventing excessive computational costs caused by overly deep partitioning trees. For evaluating its effect, we did an additional ablation study about the tree depth settings on the DBMS configuration tuning benchmark and measured **the ratio of tuning time against the total time for each tuning iteration, consisting of configuration loading, workload execution and optimization execution time** for each setting. We put this part of work in **Appendix C in the revised version**.
> > > >
> > > > The tuning time proportion can be found in the following table:
> > > >
> > > > |                     | **Optim Exec (%)** | **Config Load (%)** | **Workload Exec (%)** |
> > > > | ------------------- | ------------------ | ------------------- | --------------------- |
> > > > | **Depth=1 (TuRBO)** | 2.95 ± 1.07        | 29.16 ± 5.21        | 67.89 ± 4.87          |
> > > > | **Depth=2**         | 3.98 ± 0.41        | 28.84 ± 3.40        | 67.18 ± 3.19          |
> > > > | **Depth=3**         | 4.21 ± 0.36        | 32.44 ± 1.22        | 63.35 ± 1.22          |
> > > > | **Depth=4**         | 4.75 ± 1.04        | 32.24 ± 2.99        | 63.01 ± 3.01          |
> > > > | **Adaptive (HiBO)** | 4.31 ± 0.91        | 29.19 ± 3.28        | 66.50 ± 2.98          |
> > > >
> > > > From this table, we observe that while HiBO introduces additional computational costs during the optimization phase compared to TuRBO, the increase is minimal: even if we set the tree depth to always be 4, the extra proportion of the execution time within each iteration is **within 2% on average**. Enabling adaptive control keeps the tuning time cost of HiBO **only about 4.3% of the benchmarking time**, while it achieves the best performance improvements and S-PITR scores among all selected algorithms in this practical benchmark.
> > > >
> > > > In fact, there is no easy answer to the scalability of HiBO towards large-scale dataset especially because the search tree depth is adaptively controlled and the total computational cost can vary from different benchmarks. Further, it also depends on the types and specified configurations of clustering and classification algorithms used for search space partitioning, which can have different computational complexity. For example, SVMs with linear kernels have $O(n)$ complexity while non-linear kernels require $O(n^2)$ or $O(n^3)$, where $n$ is the number of data samples. However, the focus of this paper is the design of the hierarchical framework and we do not conduct specific analysis on designing specific component combination to large scale dataset.
> > > >
> > > > Moreover, HiBO is designed for high-dimensional tasks with limited sample budgets, which aligns with real-world scenarios where objective evaluations are expensive to be evaluated. The DBMS configuration tuning serves as a typical example because ideally we expect an important DBMS instance can keep running instead of spending time on tuning. Correspondingly, related works in the field of DBMS configuration tuning typically control their sample budget to less than 500 iterations (Zhang et al. 2021; 2022; Cereda et al. 2021; Kanellis et al. 2022). Our work even adopts the very limited sample budget (50 iterations) to form a challenging benchmark, and HiBO can improve the default performance by more than 2x within this sample budget. In fact, tuning towards opaque-box (refers to black-box, but sensitive as pointed out by Reviewer 2) problems within limited sample budget is the one of core design principles of BO and HiBO adequate exemplifies this consideration. From this perspective, scalability for large scale dataset is out of scope of this work.
> > > >
> > > > References:
> > > >
> > > > - Zhang et al. 2021 ResTune: Resource Oriented Tuning Boosted by Meta-Learning for Cloud Databases
> > > > - Zhang et al. 2022 Towards Dynamic and Safe Configuration Tuning for Cloud Databases
> > > > - Cereda et al. 2021 CGPTuner: a Contextual Gaussian Process Bandit Approach for the Automatic Tuning of IT Configurations Under Varying Workload Conditions
> > > > - Kanellis et al. 2022 LlamaTune: Sample-Efficient DBMS Configuration Tuning

---

> > > > > ### Author Response · Authors · 2024-11-22
> > > > > **Response to Reviewer 6Fsx (Thank you)**
> > > > >
> > > > > Thank you very much for your insightful suggestions and feedbacks. We hope that our responses and the revisions to the paper address your concerns and clarify any ambiguities. If you feel that these responses and changes improve the quality and clarity of the work, we kindly ask you to consider raising your score. We are happy to provide more clarification if you have additional concerns or feedbacks.

---

> > > > > > ### Comment · Reviewer_6Fsx · 2024-11-27
> > > > > >
> > > > > > I believe the author has largely addressed my concerns. I hope the revised version incorporates the issues raised in the motivation section. I have decided to increase my score.

---

> ### Author Response · Authors · 2024-11-28
> **Response to Reviewer 6Fsx (Paper Revision)**
>
> ## Motivation
>
> Thank you for carefully reviewing our responses and for increasing your score. We deeply appreciate your acknowledgment of our efforts to address your concerns and improve the paper.
>
> Following your suggestion, we have further revised the paper to explicitly incorporate the issues raised in the motivation section, ensuring clarity and alignment with your feedback:
>
> - Between **line 38 and line 47**, we included the challenges of optimization over real-world high-dimensional problems and  insufficiency of prior works in the field of DBMS configuration tuning as the example scenario. This contributes to part of our motivation for designing HiBO as an effective approach that can be applied to real-world systems (**line 70 to line 72**)
> - Between **line 58 and line 63**, we explicitly emphasize the insufficiency of prior works adopting search space partitioning for scaling BO to high-dimensional search space, describing their ineffcient utilization of generated structural information and risk of limiting exploration because of unreliable initial tree and strict sampling scope confinement.
> - Between **line 66 and line 70**, we explicitly included the motivation of designing the adaptive mechanism to control the maximum tree depth, which corresponds to the statement we put on Section 4.3 (**line 281 - line 297**)
>
> Based on the revised introduction emphasizing the motivations, the contributions are correspondingly modified (**line 70 - line 87**).
>
>
> ----------
>
>
> ## Scalability
>
> ### Pointers
>
> Besides the motivation-related contents, we additionally included an extra **Appendix G** for addressing the concerns of Scalability as suggested by Reviewer D9Dm. We discussed the scalability of this algorithm and think there can be two different interpretation of the "scalability": **the scalability with respect to increased dimensionality of the problem** and **the scalability with respect to the large scale dataset**. As the previous response provided, the scalability w.r.t large scale dataset is not commonly discussed for BO-based algorithms towards optimization within limited sample budget and the clarification is put in **Appendix G.2 (line 1120 - line 1140)**.
>
> But we further included another part of experiments on evaluating HiBO and TuRBO on Levy benchmark with increasing dimensionalities in **Appendix G.1 (line 1077 - line 1119)**, described as following.
>
> ### Result Analysis
>
> Given that the primary modification to the original local optimizer lies in the global-level navigator, our analysis focuses on comparing **the performance and timing costs of HiBO and TuRBO on the synthetic Levy benchmark for 500 iterations across varying dimensions**, as shown in **Figure 9**. Figure 9 (**Left**) depicts the ratio of mean regret (RMR) between TuRBO and HiBO ($\overline{regret}\_{TuRBO} / \overline{regret}\_{HiBO}$), while Figure 9 (**Right**) illustrates the ratio of optimization time (ROT) for HiBO relative to TuRBO ($Time_{HiBO} / Time_{TuRBO}$). Regret is defined as the difference between the optimal objective value and the value achieved by the algorithm at a given iteration, with mean regret representing the average regret over the entire optimization process. **Lower mean regret indicates better algorithm performance.**
>
> Figure 9 (Left) indicates that RMR increases steadily from ~1.03 to ~1.22 with dimensionality increasing from 50 to 1000, suggesting that TuRBO's mean regret on Levy grows more significantly than HiBO's as dimensionality rising. In other words, HiBO’s performance improvement over plain TuRBO scales effectively with increasing problem dimensions.
>
> On the other hand, as Figure 9 (**Right**) shows, although ROT remains above 1 due to the additional time cost brought by the global-level navigator of HiBO, ROT decreases from ~1.66 to ~1.18 as dimensionality increasing. This trend suggests that as the dimensionality grows, the local optimizer (i.e. TuRBO) experiences a heavier computational burden and suffers from poorer scalability compared to the global-level navigator. As a result, the relative overhead introduced by the global navigator becomes less significant as the local optimizer becomes the dominant computational component in increased dimensionalities.
>
> These results empirically demonstrate HiBO’s robust scalability with increasing dimensionality, as evidenced by its growing performance advantage over the local optimizer and the decreasing relative computational overhead of its global-level navigator.
>
> ---------------
>
> Given these updates and the addressed concerns, we would **greatly appreciate it if you could consider revisiting your score to the positive side (6 or greater)**, particularly if you feel the revisions improve the quality and clarity of the work. Thank you very much again for your evaluation and helpful feedbacks.

---

> ### Author Response · Authors · 2024-12-02
> **A Gentle Reminder for Reviewer 6Fsx**
>
> Dear Reviewer 6Fsx,
>
> Thank you once again for your valuable feedbacks and for increasing your score earlier. As indicated by previous messages sent to you, we have made further modifications to the paper following your additional guidance.
>
> As today is the last day for reviewer to post a message to the authors, we kindly ask if you might have the opportunity to review these updates. If you feel the changes sufficiently address your concerns, we would be grateful if you could consider increasing your score to the positive side.
>
> Sorry for multiple reminders sent here and we greatly appreciate your time and thoughtful contributions to this review process. Please let us know if there are any remaining questions or clarifications we can provide.
>
> Best Regards,
> The Authors

---

### Official Review · Reviewer_X5cm · 2024-11-02

**Soundness:** 3
**Presentation:** 3
**Contribution:** 3
**Rating:** 6
**Confidence:** 3

**Summary:**

The paper tackles the problem that Bayesian Optimization methods do not scale well in high-dimensional search spaces (cf. curse of dimensionality) without structure. In this context, the authors propose HiBO, a hierarchical Bayesian Optimization approach that uses global-level space partitioning information to guide the acquisition strategy of local BO modelling – which is based on the TuRBO framework of Eriksson et al. (2019) -- more effectively regarding sampling efficiency.

Several numerical experiments including synthetic benchmarks as well as database tuning problems are performed and results are compared to different baseline solutions.

**Strengths:**

S1 The considered problem is interesting and relevant.

S2 The paper is technically sound and overall well-written.

S3 The methods used are suitable.

S4 The evaluation and comparison against baselines is convincing.

**Weaknesses:**

W1 The required computational costs and scalability remain somewhat unclear.

W2 It remains unclear whether and which hyper parameters have to be tuned.

W3 Limitations of the paper’s methods could be better discussed.

**Questions:**

What are the required computational costs?

Does the approach scale?


Which hyper parameters are required (e.g. C_p, see line 263)?

How are they automatically chosen/tuned?


Minor comments:

- line 054: that integrate -> that integrates

- line 084: x and X are not defined

- line 264: What is the set for index p? Does it depend on j?

- line 316: Which sets belong to i, j, and j’?

- the references are not consistent (lower/upper case)

---

> ### Author Response · Authors · 2024-11-22
> **Response to Reviewer  X5cm (W1 & Q1, Part 1)**
>
> ## W1 & Q1
> > W1 The required computational costs and scalability remain somewhat unclear.
> >
> > Q1 What are the required computational costs? Does the approach scale?
>
> We appreciate the point about computational cost and scalability of our approach and we do understand the concern because the design of HiBO introduces the extra complexity in BO's algorithm framework.
>
> Tuning time cost is more a practical setting that we care in real-world applications. Considering this, we care more about the computational cost in DBMS configuration tuning as it provides a good testcase to observe our algorithm's practical factors. But we do not provide direct timing measurement because it will be meaningless if we do not consider **whether the cost is worth it**. Therefore, we used the custom metric S-PITR  (see Section 6.2.3) as an alternative to comprehensively measure the practical value of our algorithm, which provides the ratio of the tuning effect against the tuning time cost (and also the failure rate). As can be observed in Figure 4 (Right), HiBO achieves the greatest S-PITR score indicating the practical value even considering the tuning time cost.
>
> Specifically, HiBO incorporates an Adaptive Maximum Tree Depth mechanism to balance the exploitation and exploration while preventing excessive computational costs caused by overly deep partitioning trees. For evaluating its effect, we did an additional ablation study about the tree depth settings on the DBMS configuration tuning benchmark and measured **the ratio of tuning time against the total time for each tuning iteration, consisting of configuration loading, workload execution and optimization execution time** for each setting. We put this part of work in **Appendix C in the revised version**.
>
> The tuning time proportion can be found in the following table:
>
> |                     | **Optim Exec (%)** | **Config Load (%)** | **Workload Exec (%)** |
> | ------------------- | ------------------ | ------------------- | --------------------- |
> | **Depth=1 (TuRBO)** | 2.95 ± 1.07        | 29.16 ± 5.21        | 67.89 ± 4.87          |
> | **Depth=2**         | 3.98 ± 0.41        | 28.84 ± 3.40        | 67.18 ± 3.19          |
> | **Depth=3**         | 4.21 ± 0.36        | 32.44 ± 1.22        | 63.35 ± 1.22          |
> | **Depth=4**         | 4.75 ± 1.04        | 32.24 ± 2.99        | 63.01 ± 3.01          |
> | **Adaptive (HiBO)** | 4.31 ± 0.91        | 29.19 ± 3.28        | 66.50 ± 2.98          |
>
> From this table, we observe that while HiBO introduces additional computational costs during the optimization phase compared to TuRBO, the increase is minimal: even if we set the tree depth to always be 4, the extra proportion of the execution time within each iteration is **within 2% on average**. Enabling adaptive control keeps the tuning time cost of HiBO only about **4.3% of the benchmarking time**, while it achieves the best performance improvements and S-PITR scores among all selected algorithms in this practical benchmark.

---

> > ### Author Response · Authors · 2024-11-22
> > **Response to Reviewer X5cm (W1 & Q1, Part 2)**
> >
> > ## W1 & Q1
> > > W1 The required computational costs and scalability remain somewhat unclear.
> > >
> > > Q1 What are the required computational costs? Does the approach scale?
> >
> > As for the scalability of HiBO, there is no easy answer to the scalability of HiBO especially because the search tree depth is adaptively controlled and the total computational cost can vary from different benchmarks. Further, it also depends on the types and specified configurations of clustering and classification algorithms used for search space partitioning, which can have different computational complexity. For example, SVMs with linear kernels have $O(n)$ complexity while non-linear kernels require $O(n^2)$ or $O(n^3)$, where $n$ is the number of data samples. However, the focus of this paper is the design of the hierarchical framework and we do not conduct specific analysis on designing specific component combination to large scale dataset.
> >
> > Moreover, HiBO is designed for high-dimensional tasks with limited sample budgets, which aligns with real-world scenarios where objective evaluations are expensive to be evaluated. The DBMS configuration tuning serves as a typical example because ideally we expect an important DBMS instance can keep running instead of spending time on tuning. Correspondingly, related works in the field of DBMS configuration tuning typically control their sample budget to less than 500 iterations (Zhang et al. 2021; 2022; Cereda et al. 2021; Kanellis et al. 2022) . In fact, tuning towards opaque-box (refers to black-box, but sensitive as pointed out by Reviewer 2) problems within limited sample budget is the one of core design principles of BO. From this perspective, the scalability for large scale dataset is out of scope of this work.
> >
> > References:
> >
> > - Zhang et al. 2021 ResTune: Resource Oriented Tuning Boosted by Meta-Learning for Cloud Databases
> > - Zhang et al. 2022 Towards Dynamic and Safe Configuration Tuning for Cloud Databases
> > - Cereda et al. 2021 CGPTuner: a Contextual Gaussian Process Bandit Approach for the Automatic Tuning of IT Configurations Under Varying Workload Conditions
> > - Kanellis et al. 2022 LlamaTune: Sample-Efficient DBMS Configuration Tuning

---

> ### Author Response · Authors · 2024-11-22
> **Response to Reviewer X5cm (W2 & Q2, W3)**
>
> ## W2 & Q2
> > W2 It remains unclear whether and which hyper parameters have to be tuned.
> >
> > Q2 Which hyper parameters are required (e.g. C_p, see line 263)? How are they automatically chosen/tuned?
>
> Thanks for the question about hyperparameters. We have provided a complete list of hyperparameters with their meanings and scopes in **Appendix B** in the revised version of our paper. The explanation on the hyperparameters of HiBO include:
> -  $C_p$: Controls the balance between exploration and exploitation in the UCT calculation. Larger values increase the weight of exploration. Range: [0.5, 5.0].
> -  $\tau$: Softmax temperature for normalizing partition scores into a positive and normalized range. Range: [0.01, 2]. This hyperparameter is used for controlling the impact of global-level navigator on local optimizer and more details can be found in Appendix D. The two extreme cases of $\tau$'s value can lead to the following effects:
>     -  **Extremely high $\tau$:** Results in an overly smoothed distribution, where the difference between biases assigned to different space partitions becomes negligible. Eliminates the impact of global-level partitioning and diminishing differences between partitions;
>     - **Extremely low $\tau$**: Results in a peaky distribution and assigns overwhelming weight to the partition with the highest sampling potential, amplifying the navigator's influence.
> -  Maximum Tree Depth Limit: The limit of maximum allowable tree depth before the search process restarts after multiple times of repeated failures. It controls the tolerance of consecutive failures in one search. Range: [3, 5].
>
> We did not use advanced algorithms for tuning them but applied naive grid search for searching the best performing combinations. The parameters selected in our experiments are summarized in Appendix **A**.
>
> ## W3
> > W3 Limitations of the paper’s methods could be better discussed.
>
> Thank you for the question about limitations of HiBO may face challenges. From our experiment results and reasoning, we summarized the following two cases:
>
> 1. One limitation of HiBO lies in its relatively less pronounced improvement over the local optimizer on dense high-dimensional benchmarks compared to sparse benchmarks. While HiBO demonstrates significant performance gains on sparse benchmarks (e.g. achieving over two orders of magnitude lower regret on tasks such as Hartmann6-300D), its improvement is less evident on dense tasks where all dimensions are uniformly effective (e.g., Levy-200D, Rastrigin-200D). Our interpretation is that in sparse spaces, HiBO efficiently identifies promising regions to guide the local optimizer and it is one of its greatest advantages. While in dense spaces, it requires more samples to achieve accurate estimations across the entire space than in sparse scenarios. But HiBO still converges efficiently towards the optimal point in these tasks and achieves the best in Levy-200D and Rastrigin-200D.
> 2. HiBO may also struggle in scenarios where certain critical configuration variables are hidden or unobservable during optimization. Such situations can arise in practical tasks where parts of the configuration space are inaccessible due to hardware constraints, legacy systems, or incomplete problem specifications. In these cases, the global-level navigator may build an inaccurate model of the search space, leading to suboptimal partitioning and sampling.

---

> > ### Author Response · Authors · 2024-11-22
> > **Response to Reviewer X5cm (Minor, Thank You)**
> >
> > ## Minor
> > > - line 054: that integrate -> that integrates
> > > - line 084: x and X are not defined
> > > - line 264: What is the set for index p? Does it depend on j?
> > > - line 316: Which sets belong to i, j, and j’?
> > > - the references are not consistent (lower/upper case)
> >
> > We appreciate these "minor" but also important notes on our writing details. We have followed your advice to modify the 'that integrate' and added the explanation of $x$ and $X$ in the Preliminary part
> >
> > For $C_p$, the letter $p$ is not essentially an index but writing this parameter as $C_p$ is commonly seen in related literature such as Browne et al. 2012 and Kocsis and Szepesvári 2012. The $p$ does not mean anything special and $C_p$ as a whole forms the name of this hyperparameter.
> >
> > For the index question in the Softmax equation:
> > - firstly index $i$ does not appear in our equation and this may be a misunderstanding or a reference to a different part of the text :)
> > - On the other hand, $j$ represents the specific partition for which we are calculating its score $P_j$, while $j'$ runs over all partitions in the search space during the computation of the denominator (normalization term). This equation calculates the probability of selecting partition $j$ based on its **UCT score** ($UCT_j$) relative to the scores of all other partitions ($UCT_{j^′}$), smoothed by the temperature parameter $\tau$. The Softmax ensures the resulting partition scores are positive and sum to 1, facilitating the score to be integrated into the local optimizer's acquisition function.
> >
> > References:
> >
> > - Browne et al. 2012 A Survey of Monte Carlo Tree Search Methods
> > - Kocsis and Szepesvári 2012  Bandit Based Monte-Carlo Planning
> >
> > ---------------------
> > Thank you very much for your insightful suggestions and feedbacks. We hope that our responses and the revisions to the paper  address your concerns and clarify any ambiguities. If you feel that these responses and changes improve the quality and clarity of the work, we kindly ask you to consider raising your score. We are happy to provide more clarification if you have additional concerns or feedbacks.

---

> > > ### Comment · Reviewer_X5cm · 2024-11-26
> > >
> > > Thank you very much for your responses. Overall, I decided to keep my score.

---

### Official Review · Reviewer_q24C · 2024-11-04

**Soundness:** 3
**Presentation:** 3
**Contribution:** 3
**Rating:** 8
**Confidence:** 2

**Summary:**

This paper proposes HiBO, an algorithm that performs Bayesian optimization over high-dimensional spaces in a hierarchical manner, in which a global navigator adaptively partitions the search space into partitions with diverse sampling potential and a local Bayesian optimizer integrates global-level partitioning information to guide its acquisition strategy towards promising search space regions and hence accelerate the convergence to a local optimum. The proposal is shown to perform better than a selection of previous methods on synthetic data and is also applied to tune database management systems.

**Strengths:**

S1. Reasonable and well-argued proposal.
S2. Adaptive partitioning strategy appears to be novel in this context of exploration/exploitation.
S3. Experimentation on synthetic and real benchmarks.

**Weaknesses:**

W1. Ignores previous work on exactly the same topic of hierarchical Bayesian optimization.
W2. Ignores previous work on exactly the same application area of database tuning.
W3. Missing rationale and justification on the inclusion and exclusion of related works.

**Questions:**

The paper makes an interesting proposal and dives into the domain of adaptivity perhaps more than previous work has done.
However, the paper falls short in terms of its relationship to related work in two respects:

1. The notion of Hierarchical Bayesian Optimization Algorithm has been introduced by Pelikan and Goldberg in Hierarchical Bayesian Optimization Algorithm. Scalable Optimization via Probabilistic Modeling 2006: 63-90. The present paper does not discuss its relation to this precedent.

2. The assessment of DBMS knob tuning does not feature the most recent works in the area, by Zhang et al., even though these works are duly cited.

**Details Of Ethics Concerns:**

The paper uses the term "black box", which is advised against by ACM. "opaque" would work just as fine without alluding to stereotypes.

---

> ### Author Response · Authors · 2024-11-22
> **Response to Reviewer q24C (W1 & Q1)**
>
> ## W1 & Q1
> > W1. Ignores previous work on exactly the same topic of hierarchical Bayesian optimization.
> >
> > Q1. The notion of Hierarchical Bayesian Optimization Algorithm has been introduced by Pelikan and Goldberg in Hierarchical Bayesian Optimization Algorithm. Scalable Optimization via Probabilistic Modeling 2006: 63-90. The present paper does not discuss its relation to this precedent.
>
> Thanks for pointing out the paper published in 2006 using the same terminology 'Hierarchical Bayesian Optimization' to describe their approach. However, after carefully reading their work, we realized that their target problems are not directly related to ours: they aim to provide scalable solutions for nearly decomposable and hierarchical problems while we do not assume any decomposability or hierarchical properties of the problems to be solved. We are only concerned about the general opaque-box optimization problems especially with high dimensionality, and the "hierarchicalness" of our approach lies in the structure of the approach instead of the problem. And correspondingly, the Related Work section in our work mainly include the works about BO-based approaches in high-dimensional search space instead of those for decomposable and hierarchical problems.
>
> I hope this clarifies the rationale of our inclusion and exclusion principles about BO-related works.

---

> > ### Author Response · Authors · 2024-11-22
> > **Response to Reviewer q24C (W2 & Q2, W3)**
> >
> > ## W2 & Q2
> > > W2. Ignores previous work on exactly the same application area of database tuning.
> > >
> > > Q2. The assessment of DBMS knob tuning does not feature the most recent works in the area, by Zhang et al., even though these works are duly cited.
> >
> > Thanks for raising your concerns on the related works of DBMS configuration tuning. In fact, we provided the extra background of this benchmark itself in Section 6.2.1 and point out the drawbacks of prior BO-based works in this field such as OtterTune  (Van Aken et al. 2017) requiring large number of samples to be collected before the actual tuning, and ResTune (Zhang et al. 2021) used their pre-collected workload features and observations of 34 past tuning task prior to actual tuning. Considering the inapplicability of their pre-collected samples or meta-data and their huge tuning time cost (e.g. up to 1000 minutes for each tuning session in OtterTune), we do not directly implement or reproduce the full version of their methods in our experiments due to resource constraints. For example, for OtterTune, we only reimplemented their idea without their pre-tuning sampling workflow. The impracticality and timing cost exhibited in their work are indeed the target problems to be solved by our high-dimensional methods. As indicated by our experiments, HiBO can bring significant improvements in terms of throughputs to PostgreSQL without pre-collected data and with limited sample budget. This can form evident comparison even without directly applying their methods and provide empirical validation on the effectiveness of our approach compared with them.
> >
> > Further, those works are not highly related to the target problem we are concerned with: directly tuning over the high-dimensional configuration search space. OtterTune relies on pre-tuning sampling to choose the most important knobs for dimensionality reduction. In Zhang et al. 2021 and 2022, they manually select configuration knobs for tuning and the number of selected knobs are limited (less than 50) in their evaluation part. Choosing a limited number of knobs manually for evaluation can also be found in Cereda et al. 2022, which is another BO-based DBMS configuration tuning work.
> >
> > Instead, our method is aimed to provide a highly sample-efficient method for performance improvement based on our designed high-dimensional BO approacht, which can directly tune on the high-dimensional configuration space, without requirement for large sample budget, prior knowledge or sophisticated models. In our experiments, we only remove the configurations knobs that can directly affect experiment execution such as those related to authorization and keep most of knobs of PostgreSQL to form a high-dimensional (110-dimensional) task, without further manual selection. The most similar work to ours in this field is LlamaTune (Kanellis et al. 2022) which directly adopts high-dimensional BO method algorithms including REMBO (Wang et al. 2016) and HesBO (Nayebi et al. 2019) towards this problem without prior knowledge. We did include this approach in our evaluation following exactly the same configuration used in their paper for comparison.
> >
> > I hope this clarifies the rationale of our inclusion and exclusion principles about DBMS configuration tuning related works and our related experiment settings.
> >
> > References:
> >
> > - Van Aken et al. 2017. Automatic Database Management System Tuning Through Large-scale Machine Learning
> > - Zhang et al. 2021 ResTune: Resource Oriented Tuning Boosted by Meta-Learning for Cloud Databases
> > - Zhang et al. 2022 Towards Dynamic and Safe Configuration Tuning for Cloud Databases
> > - Cereda et al. 2022. CGPTuner: a Contextual Gaussian Process Bandit Approach for the Automatic Tuning of IT Configurations Under Varying Workload Conditions
> > - Kanellis et al. 2022. LlamaTune: Sample-Efficient DBMS Configuration Tuning
> > - Wang et al. 2016 Bayesian optimization in a billion dimensions via random embeddings
> > - Nayebi et al. 2019 A framework for Bayesian optimization in embedded subspaces

---

> > > ### Author Response · Authors · 2024-11-22
> > > **Response to Reviewer q24C (Ethical Concerns)**
> > >
> > > ## Ethical Concern
> > > > The paper uses the term "black box", which is advised against by ACM. "opaque" would work just as fine without alluding to stereotypes.
> > >
> > > We sincerely apologize for any misunderstanding caused by our use of the term "black-box." When writing our paper, we relied on the term's widespread use in computer science, particularly in the field of Bayesian Optimization, where it is a well-established concept. Besides, recently published papers (though not directly related to BO) in the ACM Digital Library (Gu et al., 2024; Xiao et al., 2024) and Nature (Tučs et al., 2024) still use this term in their titles and contents.
> > >
> > > We assure you that our usage of "black-box" was purely technical and devoid of any unintended bias. Nonetheless, we understand the importance of adhering to evolving standards in terminology. We will carefully review the relevant guidelines and consider replacing "black-box" with "opaque" or another suitable term in the revised version of our paper if deemed necessary.
> > >
> > > Thank you for highlighting this point, and we appreciate your understanding.
> > >
> > > References:
> > >
> > > - Gu et al. 2024 IsoVista: Black-Box Checking Database Isolation Guarantees
> > > - Xiao et al. 2024 Adversarial Experts Model for Black-box Domain Adaptation
> > > - Tučs et al. 2024 Extensive antibody search with whole spectrum black-box optimization

---

> > > ### Comment · Reviewer_q24C · 2024-11-27
> > > **Please clarify matters in introduction and use informative terms**
> > >
> > > Thank you for the explanation. The comment in the paper regarding this question is poorly written and does not convey the message conveyed in the above:
> > >
> > > >BO-based approaches have been proposed for tuning these configurations (Van Aken et al., 2017; Zhang et al., 2021; 2022), but they often rely on statistical methods to identify the most impactful knobs first to build a low-dimensional search space based on a large number of pre-collected samples before the actual tuning, which is costly in terms of time and computation.
> > >
> > > This comment should be instead written in the introduction, and the contrast to what is proposed in this paper should be made explicit. It should be stated that these works do not directly tune over the high-dimensional configuration search space, while the proposed method does so. To change my score, I would like to see such a clarification in the introduction and the change of the uninformative and unscientific term "black-box function" to an informative and scientific term such as "opaque function".

---

> > > > ### Author Response · Authors · 2024-11-28
> > > > **Response to Reviewer q24C (Paper Revision)**
> > > >
> > > > Thank you very much for your further reply. Considering your concerns, we have made the following adjustments in our paper:
> > > >
> > > > - As for the "hierarchicalness", we add an extra paragraph at the end of Section 2 *Background*, including the responses I provided before to explicitly clarify the indirect relevance of our paper to Pelikan et al. 2007. Please see **line 156 - line 161** for details.
> > > > - Your suggestions on explicitly mentioning the related work of DBMS configuration tuning in Introduction is helpful. We have included an extra paragraph (**between line 38 and line 46**) describing the problems of approaches in DBMS configuration tuning as part of motivations of HiBO for real-world systems. On the other hand, we clarified the details of selection of algorithms specifcially for the DBMS configuration tuning experiment in **Appendix A.3 (line 821 to line 824)** and extra explanation of LlamaTune is given in **Appendix A.2 (line 791 to line 798)**
> > > > - We carefully checked the use of "black-box" in our paper and replaced them with "opaque functions/problems", as shown in Abstract (**line 10 - line 11**), Introduction (**line 24 - line 27**) and Background (**line 98 - line 102**)
> > > >
> > > > We hope that the further revisions to the paper address your concerns and clarify any ambiguities. **If you feel that these changes do improve the quality and clarity of this paper, we kindly ask you to consider raising your score to the positive side (6 or greater).** Thank you for your helpful feedbacks and suggestions again.

---

> > > > > ### Comment · Reviewer_q24C · 2024-11-28
> > > > > **Language errors introduced in revision**
> > > > >
> > > > > Thank you for the revisions, which improve the paper.
> > > > >
> > > > > However, the same revisions have introduced some language errors, such as incomplete sentences:
> > > > >
> > > > > Sentence in Lines 063-066:
> > > > >    "While incorporating such structural information into the original BO algorithm for optimal performance introduces challenges, including designing an effective hierarchy to combine the extra module extracting structural information with the original BO, and ensuring that the structural information are utilized optimally."
> > > > >
> > > > > Sentence in Lines 158-161:
> > > > >    "While HiBO does not assume any decomposability or hierarchical properties in the optimization problems it addresses."
> > > > >
> > > > > Each of these sentences has no main clause, therefore they are incomplete.
> > > > >
> > > > > Apart from these incomplete sentences, I see the new term "hierarchicalness". That is a non-existent word, and I guess that is why it is put in quotes. It seems to me that the noun "hierarchy" already conveys well the property of being hierarchical, i.e., of using a hierarchy, so there is no need to invent a new noun for it.

---

> ### Author Response · Authors · 2024-11-22
> **Response to Reviewer q24C (Thank you)**
>
> Anyways, thank you very much for your feedbacks and pointing out the concerns. We hope that our responses and the revisions to the paper address your concerns and clarify any ambiguities. If you feel that these responses and changes improve the quality and clarity of the work, we kindly ask you to consider raising your score. We are happy to provide more clarification if you have additional concerns or feedbacks.

---

> ### Public Comment · ~Shashank_Agnihotri1 · 2024-11-22
> **Low Quality Review**
>
> Dear Everyone,
>
> This is a very poorly written review. Similar to CVPR, is ICLR planning on desk rejecting submissions from reviewers due to irresponsible reviews?
>
> Reviewer q24C mentions in their weaknesses for the paper: "Ignores previous work on exactly the same topic of hierarchical Bayesian optimization" and "Ignores previous work on exactly the same application area of database tuning" but never really bothers to cite papers or the related works that are supposedly missing! This is the worst kind of review possible! Reviewer q24C attempts to add a third weakness "Missing rationale and justification on the inclusion and exclusion of related works." which is merely a repetition of the first two mentioned weaknesses and not really a new point, again not supported by any references!
>
>
> In their questions, in Q1 the reviewer mentions a paper on a far-off topic from this submission and asks why is it not included, and as Q2 the reviewer asks why a non-peer reviewed arxiv paper that was released 4/5 months before the ICLR submission deadline not included in this submitted paper! It seems like the reviewer does not know the ICLR guidelines.
>
>
> Lastly, not only does the reviewer seem to be oblivious to ICLR guidelines, but the reviewer dares to claim "Ethical Concerns" based on some "advice" from ACM! I'm sorry but this is not an ethical concern! The use of the term "black-box" in machine learning has no roots in racism, but has roots in the history of aviation and the use of the colour black to prevent rust on flight recorders! Moreover, it is a commonly used term and never used with the intention of discrimination. These are superfluous claims at best.
>
>
> To the best of my knowledge, this is not an LLM-generated or ChatGPT-review, because, in my limited experience with LLMs (ICLR forcing down LLM-generated feedback on reviewers for reviews given to ICLR submissions this year), they would have done a better job reviewing this paper.
> As a member of the computer science community, in my humble opinion, this review should be discarded, and if possible the actions similar to CVPR could be considered for "Irresponsible Reviews".
>
> Best Regards
>
> Just a simple member of the computer science community who decided to use the "Public Comment" option in openreview

---

> > ### Comment · Program_Chairs · 2024-11-24
> >
> > The paper will be judged by its technical merits.
> >
> > ICLR has a new ethical review process this year. All accepted papers that are flagged for ethical concerns and confirmed by AC/SAC will be reviewed by a separate committee. The committee will decide whether there are ethical issues, and if so, what remedies are needed.
> >
> > Please keep all comments on OpenReview professional and polite.

---

> > > ### Comment · Reviewer_q24C · 2024-11-24
> > > **Flight recorders are painted orange**
> > >
> > > Thank you for clarifying the matter. For the record, flight recorders are painted orange and unrelated to opaque/hidden/unknown functions; such functions are not supposed to survive a computer crash, like flight recorders survive aeroplane crashes, and do not denote something undesirable either. It is better to use inclusive terms that avoid potentially offensive language and are easily understood by people who are less culturally fluent. Obtuse metaphors increase the learning curve for people being on-boarded and make it difficult to communicate with people who have not gone through that process. If a service, function, or system component is used without knowing its internals, we can call it "opaque", "hidden", or "unknown".

---

> ### Public Comment · ~Shashank_Agnihotri1 · 2024-11-26
> **Unaddressed Concerns regarding this review's quality**
>
> Dear Program Chairs,
>
> Thank you very much for the clarification on how "ethical concerns" flagged for papers would be addressed, It is indeed very helpful. I trust in the sound and informed judgment of the ACs and SACs, whatever it might be.
>
> However, the other two very important concerns regarding the review's quality remain unaddressed. I completely understand the workload of all the involved chairs and I am immensely grateful for all the efforts and work invested as they help improve the quality of the conference. Thus, I do not expect a reply, just hoping that the two issues raised in my feedback (other than the issue regarding the alleged ethical concern) on the review in "Low Quality Review" do not go unnoticed.
>
> Thank you very much!
> To keep the discussions focused on the paper itself and to avoid having to cite multiple references that show that Flight Recordings were indeed historically black and are now painted in bright orange for easy recoverability, I am not going to respond to Reviewer q24C's response.
>
> Best Regards

---

> ### Comment · Reviewer_q24C · 2024-11-27
> **This distinction of hierachicity is worth clarifying**
>
> Thank you for the reply. This distinction of different variants of hierarchicity would be good to clarify in the paper.

---

> ### Author Response · Authors · 2024-11-28
> **Response to Reviewer q24C (Language errors)**
>
> Thank you for your detailed feedback and for pointing out the language issues introduced in the revised version. We sincerely apologize for the incomplete sentences in Lines 063–066 and 158–161, as well as the use of the non-standard term "hierarchicalness."
>
> Unfortunately, the deadline for revisions has passed, and we are unable to make further edits to the submitted paper at this stage. But we acknowledge these as valid concerns and will address them in the camera-ready version if the paper is accepted. The first two errors about "while". These errors stem from our misunderstanding that "while" could function like "however" to begin an isolated sentence. We would later replace them with the following two sentences:
>
> - **However**, incorporating structural information into the original BO algorithm for optimal performance poses challenges, such as designing an effective hierarchy to integrate the module extracting structural information and ensuring the optimal utilization of this information.
> - **In the contrast**, HiBO does not assume any decomposability or hierarchical properties in the optimization problems it addresses.
>
> On the other hand, the non-existent "hierarchicalness" with double quotes is used for emphasizing the property of being hierarchical in a nuanced way, though we now recognize that the standard term "hierarchy" could have conveyed the same meaning effectively. Again, if possible, we guarantee that we would also fix this in the camera-ready version.
>
> While we acknowledge these concerns, it is worth noting that such minor issues can occasionally occur even in papers achieving 10 in ICLR. The existence of the camera-ready stage reflects the importance of providing authors with the opportunity to further refine and polish their work after revisions. We can guarantee that: **should the paper be accepted, we will ensure that all remaining language and formatting issues, including those highlighted in your comments, are thoroughly addressed in the camera-ready version.**
>
> We have worked diligently to address the major concerns raised in earlier feedbacks from you and other reviewers, and we believe these efforts do improve the paper. We hope that the quality and significance of our work will remain the primary focus of evaluation.
>
> We deeply appreciate your detailed feedbacks and hopefully you could consider revisiting your scores in light of our responses and commitment to addressing all remaining issues in the camera-ready version.

---

> > ### Comment · Reviewer_q24C · 2024-12-02
> > **Update**
> >
> > Thank you for the response. I have updated my review to reflect the above commitment.
> > Note: "In contrast" is the standard expression to be used in the second sentence.

---

> ### Author Response · Authors · 2024-12-02
> **Response to Reviewer q24C**
>
> Thank you for your reply and the extra time to re-evaluate our work. We sincerely appreciate for your decision to increase your score—it reflects the collaborative effort to refine this submission. Should the paper be accepted, we will ensure the correct usage of "In contrast" in the camera-ready version as well.
>
> Thanks again for all the detailed feedbacks you provided during this review process, which helped us a lot to improve the clarify and quality of our work.

---

### Official Review · Reviewer_wv5P · 2024-11-05

**Soundness:** 3
**Presentation:** 4
**Contribution:** 2
**Rating:** 6
**Confidence:** 3

**Summary:**

This work aims to improve the performance of approaches such as Trust Region Bayesian Optimisation (TuRBO) over higher dimensional search spaces by employing a meta algorithm that partitions the search space into more or less promising regions (in terms of containing the optimum). The main result is that the proposed meta algorithm outperforms vanilla TuRBO and other baselines (e.g., GP-based BO) for a variety of benchmarks over synthetic datasets as well as a real-world relational database tuning task. While the proposed approach appears to be more of a meta algorithm, it is empirically only explored paired with TuRBO.

**Strengths:**

S1. Originality: Simple but well-thoughtout ideas for well-motivated problem.

S2. Originality/Significance: Promising empirical results that indicate significant performance improvements over a variety of benchmarks.

S3. Clarity/Quality: Great figures illustrating main ideas and paper easy to follow; relevant works seem to be discussed and compared against.

**Weaknesses:**

W1. Significance: While the empirical results are promising, theoretical results have not been explored despite a simpler approach.

W2. Limitations: Empirical study could focus more on identifying instances where HiBO struggles.

W3. Significance: Empirical study of proposed approach limited to using TuRBO as a local optimizer.

**Questions:**

Q1. What made developing the approach challenging and what kind of theoretical properties could potentially be derived?

Q2. What are limitations of the approach and in which settings would it be expected to perform worse?

Q3. What other local optimisers could in HiBO be used instead of TuRBO and do they also get a performance boost when used with HiBO instead of just on their own?

---

> ### Author Response · Authors · 2024-11-22
> **Response to Reviewer wv5P (W1 & Q1)**
>
> ## W1 & Q1
> > W1. Significance: While the empirical results are promising, theoretical results have not been explored despite a simpler approach.
> >
> > Q1. What made developing the approach challenging and what kind of theoretical properties could potentially be derived?
>
> We appreciate raising the concern regarding the theoretical aspects of HiBO. HiBO is a dynamic and adaptive BO framework where search space partitioning is data-driven, and the search tree depth is adjusted adaptively based on recent results. These properties make deriving an accurate theoretical analysis challenging, but we can provide a simplified analysis showing how HiBO mitigates over-exploration in vanilla GP-based BO.
>
> HiBO employs a partition-score-guided acquisition strategy that balances exploration and exploitation at a finer granularity. Specifically, the global-level navigator partitions the search space into partitions $\{\Omega_1, \Omega_2, \dots, \Omega_k\}$ and estimates the distribution of sampling potential over these partitions, assigning each partition $\Omega_j$ a score $P_j$ based on Upper Confidence bound for Trees (UCT). Assuming the clustering and classification algorithms perform adequately, partition scores of different partitions are expect to exhibit great difference, and the most promising partition $\Omega^*$ containing the global optimum $x^*$ should be assigned the greatest partition score such that $P^* \geq P_j, \space \forall \Omega^* \neq \Omega_j$
>
> On the other hand, the local BO-based optimizer integrates these scores into its acquisition function:
>
> $\alpha_P(x; D_t) = \alpha(x; D_t) \cdot P_{\text{leaf}(x)}$
>
> where $\alpha(x; \mathcal{D}_t)$ is the acquisition function used in standard BO, $D_t$ is the dataset accumulated til the current $t$-th iteration, and $leaf(x)$ refers to the partition containing $x$. This approach biases sampling toward high-potential space partitions while preserving exploration in under-sampled areas.
>
> Vanilla GP-based BO often suffers from over-exploration due to its reliance on uncertainty $\sigma(x)$ in acquisition functions, particularly in high-dimensional spaces. For example, in the UCB acquisition function defined as below:
>
> $\alpha_{\text{UCB}}(x; D_t) = \mu_t(x) + \kappa \sigma_t(x)$
>
> where $\mu_t(x)$ is the estimated value of objective function at $x$, $\sigma_t(x)$ is the corresponding predicted standard deviation at $x$   and $\kappa$ is a constant controlling the exploration-exploitation trade-off. Regions with high uncertainty $\sigma_t(x)$ dominate this value. This bias causes excessive sampling in uncertain but low-potential regions, exacerbated by the curse of dimensionality, where sparse data leads to uniformly high uncertainties.
>
> Compared to vanilla GP-based BO, HiBO integrates the global-level sampling potential distribution into this acquisition function. Given partition scores are differentiated by their sampling potentials, acquisition function values of samples $x_{low}$ from low-potential regions will be degraded by weighing them with low partition score $P_{\text{leaf}(x)}$. While samples from promising regions can have amplified acquisition function values. As mentioned, when the clustering and classification algorithms perform well, the acquisition function values of samples of most promising partitions $P^*$ can be greatly enhanced, which guides the search towards more promising regions.
>
> A challenging factor for developing this approach lies in how to control the depth of the constructed search tree. At early optimization stages, when the dataset is limited, overly deep partitions may prioritize overly specific but suboptimal regions. Conversely, a constant shallow tree underutilizes the dataset, resulting in insufficient exploitation. To address this, we designed an adaptive mechanism to adjust the maximum tree depth based on recent evaluation results, balancing exploitation, exploration, and computational cost. While this adaptive mechanism complicates theoretical analysis, it empirically enhances HiBO’s performance and efficiency. Please see Appendix **C** for a more detailed explanation on its role.

---

> ### Author Response · Authors · 2024-11-22
> **Response to Reviewer wv5P (W2 & Q2, W3 & Q3, Thank you)**
>
> ## W2 & Q2
> > W2. Limitations: Empirical study could focus more on identifying instances where HiBO struggles.
> >
> > Q2. What are limitations of the approach and in which settings would it be expected to perform worse?
>
> Thank you for the question about limitations and scenarios where HiBO may face challenges. From our experiment results and reasoning, we summarized the following two cases:
> 1. One limitation of HiBO lies in its relatively less pronounced improvement over the local optimizer on dense high-dimensional benchmarks compared to sparse benchmarks. While HiBO demonstrates significant performance gains on sparse benchmarks (e.g. achieving over two orders of magnitude lower regret on tasks such as Hartmann6-300D), its improvement is less evident on dense tasks where all dimensions are uniformly effective (e.g., Levy-200D, Rastrigin-200D). Our interpretation is that in sparse spaces, HiBO efficiently identifies promising regions to guide the local optimizer and it is one of its greatest advantages. While in dense spaces, it requires more samples to achieve accurate estimations across the entire space than in sparse scenarios. But HiBO still converges efficiently towards the optimal point in these tasks and achieves the best in Levy-200D and Rastrigin-200D.
> 2. HiBO may also struggle in scenarios where certain critical configuration variables are hidden or unobservable during optimization. Such situations can arise in practical tasks where parts of the configuration space are inaccessible due to hardware constraints, legacy systems, or incomplete problem specifications. In these cases, the global-level navigator may build an inaccurate model of the search space, leading to suboptimal partitioning and sampling.
>
>
> ## W3 & Q3
> > W3. Significance: Empirical study of proposed approach limited to using TuRBO as a local optimizer.
> >
> > Q3. What other local optimisers could in HiBO be used instead of TuRBO and do they also get a performance boost when used with HiBO instead of just on their own?
>
> It is good point about the generalization of our algorithm to guide local optimizers based on other algorithms, and sorry for missing this part when submitting the initial version. Considering almost all BO-based optimization algorithms have their acquisition-function-based strategy for selecting next data samples, the hierarchical feature and the acquisition-based augmentation adopted by HiBO can be applied to any other local optimizer. The choice of TuRBO was inspired by its effectiveness and simplicity for implementation.
>
> Considering this point, we just conducted additional experiments to measure the effectiveness of applying HiBO for guiding the vanilla Gaussian-Process-based BO in synthetic benchmarks and the newly added real-world benchmarks (Mopta08 and Rover Trajectory Planning, see Appendix **E** in the revised paper). The measurements can be found as the additional **brown** curve named 'HiBO-GP' and the effectiveness of HiBO can be reflected by the gap between it and the curve for vanilla GP-based BO ('GP', blue curve). Among all the synthetic benchmarks, HiBO-GP consistently achieves better results than vanilla GP though to different extents. It can even achieve the second best regret value in 500-dimensional Branin2 task. In the two newly-added real-world benchmarks, HiBO-GP achieves more than 25% less vehicle mass for Mopta08 and about 2 more reward values than vanilla GP-BO on average, further indicating the effectiveness of our design.
>
>
> -----------------------------
> Thank you very much for your insightful  suggestions. We hope that our responses and the revisions to the paper adequately address your concerns and clarify any ambiguities. If you feel that these changes improve the quality and clarity of the work, we kindly ask you to consider raising your score. We are happy to provide more clarification if you have additional concerns or feedbacks.

---

> > ### Comment · Reviewer_wv5P · 2024-11-26
> >
> > Thank you, these are very insightful answers and cover the questions I had!

---

### Comment · Reviewer_6Fsx · 2024-11-27

[review text omitted: it was posted to a different submission]

---

> ### Author Response · Authors · 2024-11-28
> **Potentially incorrect reviews**
>
> Thank you for your engagement and comments on this page. However, it seems this part of your feedback may have been mistakenly associated with our paper, as it references content that appears unrelated to the submission under discussion of this page
>
> Could you kindly verify whether this comment was intended for another paper? If so, it may help to repost your feedback to the appropriate page to ensure it reaches the correct authors :)

---

### Meta-Review · Area_Chair_YKK9 · 2024-12-16

**Metareview:**

This paper addresses the challenge of high-dimensional Bayesian optimization by proposing HiBO, a hierarchical approach that adaptively partitions the search space to guide the optimization process. HiBO combines a global navigator for partitioning with a local Bayesian optimizer (based on TuRBO) that leverages partitioning information to focus on promising regions. The authors demonstrate empirical improvements over existing methods, including TuRBO and GP-based BO, on synthetic benchmarks and a real-world database tuning task.

However, reviewers also identify some weaknesses:

- Lack of Theoretical Analysis: The paper primarily relies on empirical evaluation. A theoretical analysis of HiBO's properties, such as convergence guarantees or regret bounds, would significantly strengthen the work.
- Limited Scalability Discussion: While the paper aims to address high-dimensional optimization, a more in-depth discussion of scalability is needed.

Overall, the paper is interested but below the acceptance bar.

**Additional Comments On Reviewer Discussion:**

I carefully read the discussion and the comments. The authors asked to not take into account Reviewer q24c initially(then they change their opinion when reviewer q24c raised the score), the reviewer also got some external pressure to change the review that was not too strongly motivated in my opinion as a result I decided to discount reviewer q24c final rating

---

### Decision · Program_Chairs · 2025-01-22

Reject